# Multi-Agent Reinforcement Learning with Submodular Reward

Wenjing Chen [1]   Chengyuan Qian [1]   Shuo Xing [1]   Yi Zhou [1]   Victoria G. Crawford [1]

## Abstract

In this paper, we study cooperative multi-agent reinforcement learning (MARL) where the joint reward exhibits submodularity, which is a natural property capturing diminishing marginal returns when adding agents to a team. Unlike standard MARL with additive rewards, submodular rewards model realistic scenarios where agent contributions overlap (e.g., multi-drone surveillance, collaborative exploration). We provide the first formal framework for this setting and develop algorithms with provable guarantees on sample efficiency and regret bound. For known dynamics, our greedy policy optimization achieves a $1/2$-approximation with polynomial complexity in the number of agents $K$, overcoming the exponential curse of dimensionality inherent in joint policy optimization. For unknown dynamics, we propose a UCB-based learning algorithm achieving a $1/2$-regret of $O(H^2 K S \sqrt{AT})$ over $T$ episodes.

## 1. Introduction

Cooperative multi-agent reinforcement learning (MARL) has emerged as a powerful paradigm for coordinating teams of interactive agents in complex environments, with applications ranging from robotic systems (Hüttenrauch et al., 2017; Ismail et al., 2018) to network optimization (Zhang and Lesser, 2011) and resource allocation (Rauch et al., 2024; Tilahun et al., 2023). In cooperative MARL, agents coordinate their actions in a stochastic environment to maximize the expected cumulative reward, which is formalized as a multi-agent Markov decision process (MDP).

A common simplifying assumption is that the joint reward observed at each time step is a *linear* (additive) function of individual agent contributions (Li et al., 2021; Hsu et al., 2024). While this assumption facilitates decomposition and

analysis, it fundamentally limits the expressiveness of the reward model. In many real-world collaborative tasks, an agent's contribution to the global objective depends critically on the actions of other agents, leading to interaction effects such as overlap, redundancy, or saturation that cannot be adequately captured by additive reward structures.

Such phenomena arise naturally in a variety of collaborative decision-making problems. In multi-robot map-exploration, for instance, a team of robots explores an unknown environment to build a comprehensive map with the goal of maximizing information gain (Ma et al., 2020). The information gained by each robot may overlap with that of others and exhibits a diminishing returns property as more robots are deployed to the search task. Similarly, in multi-drone surveillance and tracking, a team of drones monitors a large geographic area with the objective of maximizing the number of distinct objects under surveillance (Zhang and Lesser, 2011). Here, the coverage achieved by different drones often overlaps, again resulting in diminishing returns as more agents are assigned to the task. Ignoring such overlap effects can cause agents to converge to redundant behaviors, preventing efficient exploration and undermining effective cooperation (see Figure 1).

To model such settings, we study a natural class of cooperative MARL problems where the joint reward exhibits *submodularity*, which is a fundamental property from combinatorial optimization that captures diminishing marginal returns (Hassidim and Singer, 2017; Iyer et al., 2021). Submodular functions are ubiquitous in machine learning and optimization (Chen et al., 2025a; Yue and Guestrin, 2011), encompassing important cases such as set cover, entropy, mutual information, and facility location objectives. This property naturally describes cooperative MARL tasks where agents' contributions overlap, the diversity of the agents' behavior is important, or where coverage gains saturate, and the marginal benefit of adding an agent to a smaller team is at least as large as adding it to a larger team (Li et al., 2021). Notably, the coverage objective in multi-drone surveillance is monotone submodular (Zhang and Lesser, 2011), and information gain in collaborative exploration can be quantified via entropy or mutual information, both of which are submodular functions (Ma et al., 2020).

Motivated by these considerations, we introduce Multi-

[1]Department of Computer Science and Engineering, Texas A&M University. Correspondence to: Victoria G. Crawford <vcrawford@tamu.edu>.

*Proceedings of the 43ʳᵈ International Conference on Machine Learning*, Seoul, South Korea. PMLR 306, 2026. Copyright 2026 by the author(s).

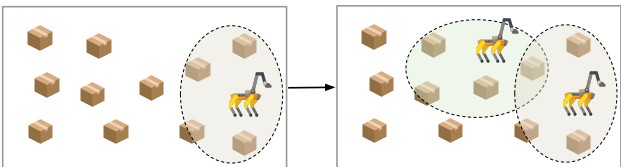

*Figure 1.* Demonstration of multi-agent collaboration.

Agent Reinforcement Learning with Submodular Rewards (MARLS), a novel cooperative MARL framework in which agents jointly influence a monotone submodular reward function at each time step. We consider an episodic setting with shared state and action spaces, where the goal is to maximize the total expected cumulative return. Despite the structural regularity imposed by monotonicity and submodularity, MARLS presents significant algorithmic and computational challenges. First, even with the monotonicity and submodularity assumptions, finding optimal policies in MARLS remains computationally intractable: As we illustrate in Section 4.1.1, the single-step case reduces to the NP-hard problem of submodular maximization under partition matroid constraints. Second, although the classical Bellman optimality equation (Lemma 2) (Bellman, 1957; Sutton et al., 1998) provides a recursive characterization of optimal policies, directly evaluating the value function of a given stochastic policy requires time and memory exponential in the number of agents $K$. Finally, while submodular maximization has been extensively studied in static settings, extending it to sequential decision-making settings introduces substantial difficulties stemming from stochastic transition dynamics and the dependence of each agent's marginal reward contribution on the policies and randomness of previously optimized agents. This is further amplified when the transition dynamics are unknown.

In particular, the contributions of the paper are as follows:

1. We propose the Multi-Agent Reinforcement Learning with Submodular Rewards (MARLS) setting in Section 3.1. We characterize the inherent computational challenges of MARL and establish that even the single-step MARLS problem is NP-hard (Section 4.1.1). To address these challenges, we consider the factorized policies and propose the marginal value decomposition method. This motivates our development of tractable approximation algorithms for MARLS.

2. We propose `Greedy Policy Optimization` in Section 4.2 for the case where the transition dynamics are known. `Greedy Policy Optimization` leverages the monotonicity and submodularity of the reward function as a tractable solution for MARLS. Our key insight is that the submodular structure enables efficient greedy sequential optimization to find a *decomposable* policy, which admits polynomial-size representations, and enables distributed execution (see Section 4.1.2). Despite this structural restriction, we prove

that `Greedy Policy Optimization` achieves a 1/2 -approximation guarantee with respect to the optimal (potentially non-decomposable) joint policy.

3. For the case where the transition dynamics are unknown, in Section 4.3, we present and analyze an upper confidence bound-based reinforcement learning algorithm, `UCB-GVI`, that combines optimistic exploration with greedy submodular maximization. `UCB-GVI` achieves a $\frac{1}{2}$-approximation regret of $O(S^2 A H^3 K^2 \log T + H^2 K S \sqrt{AT})$ over $T$ episodes (Theorem 2), which is the first sublinear regret guarantee for MARLS. We present the major technical novelty and contributions of our analysis in Section 4.3.

## 2. Related Work

### 2.1. Multi-Agent Reinforcement Learning

A fundamental challenge in cooperative MARL is the exponential growth of the joint state-action space with the number of agents $K$, which often renders traditional single-agent RL methods impractical. To address such complexity, decentralized policies are typically employed (Sukhbaatar et al., 2016; Foerster et al., 2016; Chen et al., 2022), where each agent selects its action conditioned only on its local action-observation history. While policy execution is decentralized, training may be centralized. Decentralized approaches focus on communication efficiency and learning coordination protocols. Another line of work addresses this scalability challenge through value decomposition methods, which impose structural constraints on the joint action-value function. (Sunehag et al., 2017; Rashid et al., 2020; Ma et al., 2021; Dou et al., 2022) address scalability by imposing structural constraints such that local optima on per-agent action-values correspond to the global optimum on the joint action-value.

Some existing work in multi-agent reinforcement learning has addressed the need to capture dependency structures in real-world collaborative tasks by encouraging diverse behavior between agents (Mahajan et al., 2019; Wang et al., 2020; Li et al., 2021; Jiang and Lu, 2021). Li et al. proposed a novel information-theoretic objective based on the mutual information between agents' identities and trajectories, which is a special case of ours. However, most existing methods implicitly assume that agent contributions combine in simple ways, either additively or through monotonic mixing, and do not exploit the intrinsic structure of reward functions that exhibit submodularity.

### 2.2. Submodular Maximization under Matroid Constraint

Monotone submodular maximization with a matroid constraint (MSMM) is an NP-hard combinatorial optimization

problem[1] and represents one of the most extensively studied problems in submodular optimization (Feldman et al., 2025; Calinescu et al., 2011; Banihashem et al., 2024). Given the computational complexity of MSMM, research efforts have focused on developing and analyzing approximation algorithms. The standard greedy algorithm achieves a $\frac{1}{2}$-approximation ratio (Fisher et al., 1978), while more sophisticated algorithms using continuous extensions (multilinear relaxations) attain a $\left(1 - \frac{1}{e}\right)$-approximation (Calinescu et al., 2007; Badanidiyuru and Vondrák, 2014). However, these latter methods typically incur significantly higher query complexity than the comparatively simple greedy approach. As detailed in Section 3, the inherent difficulty of MSMM contributes substantially to the computational challenges of the setting considered in this work. Specifically, our problem reduces to MSMM when the Markov decision process is restricted to a single step.

Another related line of work concerns combinatorial multi-armed bandits with submodular reward structures. A detailed discussion is provided in Appendix A.

# 3. Preliminaries

In this section, we introduce the background and formulation of Multi-Agent Reinforcement Learning with Submodular Rewards (MARLS). We begin with key notation and then present the core problem formulation.

## 3.1. Multi-Agent MDP with Submodular Reward

We consider a cooperative episodic multi-agent reinforcement learning setting with $K$ agents. The environment is modeled as a tabular Markov Decision Process (MDP) where agents interact over $H$ discrete time steps per episode. Agents share the same local state space $\mathcal{S}$ and action space $\mathcal{A}$. Their interaction is modeled by a Multi-Agent Markov Decision Process with a *submodular* global reward function. Formally, we define a Multi-Agent MDP with Submodular Reward (MAMDP-SR) as follows:

**Definition 1** (Multi-Agent Markov Decision Processes with Submodular Reward (MAMDP-SR)). *MAMDP-SR is characterized by a tuple $M = (\mathcal{S}, \mathcal{A}, \{P_{i,h}\}_{i \in [K], h \in [H]}, r, K, H)$ where:*

1. *$H$ is the episode length,*
2. *$K$ is the number of agents,*
3. *$\mathcal{S}$ is a finite state space common to all agents,*
4. *$\mathcal{A}$ is a finite action space common to all agents,*
5. *$P_{i,h} : \mathcal{S} \times \mathcal{A} \times \mathcal{S} \to [0, 1]$ is the transition probability function for agent $i$ at time step $h$, with $P_{i,h}(s' \mid s, a)$ denoting the probability that agent $i$ transitions to state $s'$ after taking action $a$ in state $s$ at step $h$.*

---

[1]MSMM is proven to be NP-hard unless NP has $n^{\mathcal{O}(\log(\log(n)))}$-time deterministic algorithms (Wolsey, 1982; Feige, 1998).

6. *$r : (\mathcal{S} \times \mathcal{A})^K \to [0, 1]$ is the global reward function dependent on the joint state space and action space of all agents. Further, $r$ is assumed to be monotone and submodular (see Section 3.3).*

## 3.2. Episode Dynamics

At the start of an episode ($h = 1$), the agents are initialized at joint state

$$\bar{\mathbf{s}}_1 = (\bar{s}_1^1, \ldots, \bar{s}_1^K).$$

Then at each time step $h \in [H]$, each agent $i \in [K]$ observes its state $s_h^i \in \mathcal{S}$, and chooses an action $a_h^i$ from the action space $\mathcal{A}$. Let us denote the joint state vector and action vector as $\mathbf{s}_h = (s_h^1, \ldots, s_h^K)$ and $\mathbf{a}_h = (a_h^1, \ldots, a_h^K)$ where $s_h^i \in \mathcal{S}$ is the state of the $i$-th agent, and $a_h^i$ is the action chosen by the agent $i$ at time step $h$. The environment then emits a global reward $r(\mathbf{s}_h, \mathbf{a}_h)$ and each agent transitions independently to its next state $s_{h+1}^i$ which is drawn from the distribution $P_{i,h}(\cdot | s_h^i, a_h^i)$ at time step $h + 1$. For notation simplicity, we use $P_h(\mathbf{s}_{h+1} | \mathbf{s}_h, \mathbf{a}_h)$ to denote the probability of transition to joint state $\mathbf{s}_{h+1}$ from joint state-action pair $(\mathbf{s}_h, \mathbf{a}_h)$ at time step $h$, and $P$ to denote the overall transition model of all the agents at all time steps. We consider the setting where the transition dynamics of each agent are independent. i.e., $P_h(\mathbf{s}_{h+1} | \mathbf{s}_h, \mathbf{a}_h) = \prod_{i=1}^K P_{i,h}(s_{h+1}^i | s_h^i, a_h^i)$ where $s_h^i \in \mathcal{S}$ is the state of the $i$-th agent, and $a_h^i$ is the action chosen by the agent $i$ at time step $h$.

A (stochastic) policy at step $h$ is a mapping $\pi_h : \mathcal{S}^K \times \mathcal{A}^K \to [0, 1]$ where $\pi_h(\mathbf{s}, \mathbf{a})$ is the probability of taking joint action $\mathbf{a}$ in joint state $\mathbf{s}$. It then follows that $\sum_{\mathbf{a}:\mathbf{a} \in \mathcal{A}^K} \pi_h(\mathbf{s}, \mathbf{a}) = 1$ for any $h \in [H]$ and $\mathbf{s} \in \mathcal{S}^K$. For notation simplicity, we use $\pi := \{\pi_h\}_{h \in [H]}$ to denote the policy over all time steps.

## 3.3. Submodular Rewards

The key characteristic of our model is that the global reward is not additive across agents, but instead reflects diminishing returns—a property captured by submodularity. Formally, we assume the reward is determined by a monotone submodular set function $f : 2^{\mathcal{S} \times \mathcal{A}} \to [0, 1]$ evaluated on the set of current agent state-action pairs:

$$r(\mathbf{s}, \mathbf{a}) = f\left(\{(s^1, a^1), (s^2, a^2), \ldots, (s^K, a^K)\}\right). \quad (1)$$

where $\mathbf{s} = (s^1, s^2, ..., s^K)$ is the joint state, and $\mathbf{a} = (a^1, a^2, ..., a^K)$ is the joint action. Here we use the notation $\{x^1, x^2, ...x^n\}$ to denote the set of the elements of $x^1, x^2, ..., x^n$. The definition indicates that the reward is determined by the state-action pair of each agent. Furthermore, here we assume that there exists an oracle access $f$ for any subset of state-action pairs. i.e., for any $S \in 2^{\mathcal{S} \times \mathcal{A}}$, we have oracle access to $f(S)$.

**Assumption 1** (Monotone Submodular Reward). *We assume that for any subsets of the state-action pairs $A \subseteq B \subseteq 2^{S \times A}$, and state-action pair $x$ such that $x \in 2^{S \times A}/B$, it follows that*

$$f(A \cup \{x\}) - f(A) \geq f(B \cup \{x\}) - f(B)$$
$$f(A) \leq f(B).$$

We define the marginal gain of adding a new state-action pair $x$ to a state-action pair set $A$ as $\Delta f(A, x)$, i.e., $\Delta f(A, x) = f(A \cup x) - f(A)$.

Next, we define the objective of the problem, which is to maximize the total reward obtained in an episode. To do that, we define value function $V$ and $Q$ as follows. $V_h^\pi(\mathbf{s}, \mathbf{a})$ is defined as the expected total reward from step $h$ to the final time step $H$ given that $\mathbf{s}_h = \mathbf{s}$, and $\mathbf{a}_h = \mathbf{a}$.

$$V_h^\pi(\mathbf{s}) = \mathbb{E}_\pi \Big[ \sum_{t=h}^{H} r(\mathbf{s}_t, \mathbf{a}_t) | \mathbf{s}_h = \mathbf{s} \Big].$$

The action-value function $Q_h^\pi : \mathcal{S}^K \times \mathcal{A}^K \to \mathbb{R}$ is defined analogously conditioned on taking joint action $\mathbf{a}_h = \mathbf{a}$:

$$Q_h^\pi(\mathbf{s}, \mathbf{a}) := \mathbb{E}_\pi \Big[ \sum_{t=h}^{H} r(\mathbf{s}_t, \mathbf{a}_t) \Big| \mathbf{s}_h = \mathbf{s}, \mathbf{a}_h = \mathbf{a} \Big].$$

The goal is to find an optimal policy $\pi^*$ that maximizes the expected total reward from the initial state:

$$\pi^* \in \operatorname{argmax}_\pi V_1^\pi(\bar{\mathbf{s}}_1).$$

Next, we illustrate the above problem setup in the example of multi-robot coordination and tracking (Xu et al., 2023). In this setting, a team of $K$ drones operates cooperatively, with each drone modeled as an individual agent. The object is to determine a global strategy for these drones to capture as many objects as possible. The state space $\mathcal{S}$ is the spatial location of the drones, and the action space $\mathcal{A}$ specifies the movement actions available to each drone. Therefore, the transition dynamics of different drones are independent. The reward at time $t$ is the number of distinct objects covered by the drone team. If $M$ objects exist, we can define $r_t(\mathbf{s}, \mathbf{a}) = \sum_{i=1}^{M} \mathbb{I}(\text{object } i \text{ is captured by the agents in state } \mathbf{s})$, where $\mathbb{I}(\text{object } i \text{ is captured by the agents in state } \mathbf{s})$ is an indicator function about whether the object $i$ is covered by the drones in the joint state $\mathbf{s}$. This is a submodular set-cover function bounded by $[0, M]$.[2]

## 4. Main Results

In this section, we present our main algorithmic and theoretical contributions for MARLS. We begin by establishing

---

[2]In this paper, we assume the function $f$ to be bounded in $[0, 1]$, and our results can be extended to the $[0, M]$ case by scaling.

the computational barriers inherent in the problem, then propose tractable solutions based on sequential greedy policy optimization that leverage submodularity under two settings: the planning setting (known transition dynamics) and the online learning setting (unknown transition dynamics).

### 4.1. Computational Challenges and Value Function Decomposition

#### 4.1.1. COMPUTATIONAL CHALLENGES

The classical Bellman optimality equation for multi-agent MDPs provides a recursive characterization of general and optimal policies (See Lemma 2 in Appendix B.1). However, directly solving the Bellman equation and the optimal policy, as is formulated in Equation (3) and (4) from Lemma 2, are computationally infeasible: evaluating the value functions $V$ and $Q$ of any given stochastic policy $\pi$ as in (3) requires $O(|\mathcal{S}|^{2K}|\mathcal{A}|^K H)$ time and $O(|\mathcal{S}|^K|\mathcal{A}|^K H)$ memory, both exponential in the number of agents $K$. This exponential complexity arises from two fundamental challenges:

1. A general joint policy $\pi = \{\pi_h(\mathbf{s}, \mathbf{a})\}_{h \in [H]}$ requires $O(|\mathcal{S}|^K|\mathcal{A}|^K H)$ space to store, which is exponential in $K$.

2. The Bellman equation update also requires computation and memory complexities that are exponential to the agent number $K$.

To circumvent this curse of dimensionality, we exploit the submodularity structure of the reward function. Our goal is to propose an algorithm that outputs a policy with high expected total reward that runs in computation complexity and memory complexity both polynomial in $K$.

However, below we explain that even with the submodularity assumption, finding optimal policies in MARLS with polynomial complexity in $K$ remains computationally intractable. To illustrate this, consider the single-step case ($H = 1$). The value function reduces to

$$Q_1^\pi(\mathbf{s}, \mathbf{a}) = \mathbb{E}_\pi[r(\mathbf{s}_1, \mathbf{a}_1) | \mathbf{s}_1 = \mathbf{s}, \mathbf{a}_1 = \mathbf{a}]$$
$$= r(\mathbf{s}, \mathbf{a}).$$

By the Bellman optimality equation, the optimal policy satisfies

$$\pi_1^*(\bar{\mathbf{s}}_1) = \arg \max_{\mathbf{a} \in \mathcal{A}^K} r(\bar{\mathbf{s}}_1, \mathbf{a})$$
$$= \arg \max_{a^1, \dots, a^K} f\big(\{(\bar{s}^1, a^1), \dots, (\bar{s}^K, a^K)\}\big).$$

This is equivalent to submodular maximization under a partition matroid constraint, where each agent's action space forms a partition and the constraint requires selecting at most one action per partition (see Appendix B.2). Since partition

matroids are a special case of general matroid constraints, and submodular maximization under matroid constraints is NP-hard unless NP $\subseteq n^{\mathcal{O}(\log \log n)}$-time (Feige, 1998), finding optimal policies in MARLS is computationally intractable even for $H = 1$.

Fortunately, the classical greedy algorithm for monotone submodular maximization achieves a $1/2$-approximation (Wolsey, 1982). Motivated by this, we develop a greedy sequential policy optimization approach that leverages submodular structure to achieve polynomial complexity with provable approximation guarantees.

### 4.1.2. JOINT POLICIES AND REWARD DECOMPOSITION

To overcome the challenge of the exponential memory complexity as illustrated in 1, we define another class of policies that can be factorized and decomposed into local agent policies, which are also referred to as decomposable policies:

**Definition 2.** *A policy $\pi$ is called decomposable i.f.f. there exists local policies $\{\pi_{i,h}\}_{i\in[K],h\in[H]}$ such that*

$$\pi_h(\mathbf{s}, \mathbf{a}) = \prod_i \pi_{i,h}(s^i, a^i).$$

*Here $\pi_{i,h}(s, a)$ is the probability of the $i$-th agent choosing action $a$ given state $s$ at time step $h$, with $\sum_a \pi_{i,h}(s, a) = 1$ for any $i \in [K]$, $h \in [H]$ and $s \in \mathcal{S}$.*

Following the previous convention, we use $\pi_i := \{\pi_{i,h}\}_{h\in[H]}$ to denote the $i$-th agent's policy over all time steps.[3] The joint policies of the agents are indexed by $[i] = \{1, \ldots, i\}$ as $\pi_{[i]}$. More specifically, $\pi_{[i],h}(s^{[i]}, a^{[i]}) = \prod_{l=1}^i \pi_{l,h}(s^l, a^l)$, where $s^{[i]} = (s^1, s^2, ..., s^i), a^{[i]} = (a^1, a^2, ...a^i)$ denotes the partial state vector and action vector. Notice that decomposable policies admit polynomial-size representations (requiring only $O(K|\mathcal{S}||\mathcal{A}|H)$ memory) and enable distributed execution, as each agent can independently select actions based on its local state. However, restricting to decomposable policies introduces approximation error, since the optimal policy $\pi^*$ may not be decomposable.

To tractably optimize the global reward, we decompose the submodular function $f$ into marginal contributions. Define the partial reward induced from the agents in the index set $[i]$ for the state $\mathbf{s}$ and action $\mathbf{a}$ as

$$r(\mathbf{s}, \mathbf{a}; [i]) \triangleq f(\{(s^l, a^l)\}_{l\in[i]}),$$

and the marginal gain of adding agent $i$ as

$$\Delta r_i(\mathbf{s}, \mathbf{a}) \triangleq r(\mathbf{s}, \mathbf{a}; [i]) - r(\mathbf{s}, \mathbf{a}; [i-1]).$$

---

[3]Note that $\pi_i$ and $\pi_h$ may appear in similar forms. The subscript distinguishes their meaning: letters $i$ or $j$ denote an agent's policy over all steps, and $h$ denote the joint policy at a certain step.

Then, by telescoping, the total reward decomposes as

$$r(\mathbf{s}, \mathbf{a}) = \sum_{i=1}^K \Delta r_i(\mathbf{s}, \mathbf{a}).$$

### 4.1.3. MARGINAL VALUE AND $Q$-FUNCTIONS

Given fixed policies $\pi_{[i-1]} = \{\pi_{j,h}\}_{j\in[i-1],h\in[H]}$ for the first $i-1$ agents and the initial state $\bar{\mathbf{s}}_1$, the state-action trajectory distribution $\{(s_h^j, a_h^j)\}_{h\in[H],j\in[i-1]}$ for the first $i-1$ agents is determined. Consequently, the expected marginal gain from adding the $i$-th agent is also fixed. From this perspective, agent $i$ can be viewed as operating in an environment induced by the fixed behavior of agents $1, \ldots, i-1$, where its reward corresponds to the marginal gain it contributes relative to the existing team. Thus, fixing the policies of the first $i-1$ agents induces a well-defined reward function for the $i$-th agent. For a given policy $\pi_{[i-1]}$, we define the *expected marginal reward* for agent $i$ executing action $a$ at state $s$ and step $h$ as

$$\begin{aligned} &R_{i,h}(s, a \mid \pi_{[i-1]}) \\ &= \mathbb{E}\Big[\Delta r_i(\mathbf{s}_h, \mathbf{a}_h)\Big| s_h^i = s, a_h^i = a, \pi_{[i-1]}\Big], \quad (2)\end{aligned}$$

where the expectation is taken over the distribution of the first $i-1$ agents' state-action pairs $(s_h^{[i-1]}, a_h^{[i-1]})$ induced by $\pi_{[i-1]}$. For fixed policies $\pi_{[i-1]}$ of the first $i-1$ agents, we define agent $i$'s *marginal value function* with policy $\pi_i$ as:

$$V_h^{\pi_i}(s; \pi_{[i-1]}) = \mathbb{E}_{\pi_i}\Big[\sum_{t=h}^H \Delta r_i(\mathbf{s}_t, \mathbf{a}_t)|s_h^i = s, \pi_{[i-1]}\Big]$$

$$Q_h^{\pi_i}(s, a; \pi_{[i-1]})$$
$$= \mathbb{E}_{\pi_i}\Big[\sum_{t=h}^H \Delta r_i(\mathbf{s}_t, \mathbf{a}_t)|s_h^i = s, a_h^i = a, \pi_{[i-1]}\Big].$$

These marginal value functions measure the expected total marginal reward contributed by agent $i$ when added to the team $[i-1]$. Critically, the following lemma demonstrates that once the policies of preceding agents are fixed, agent $i$ faces a single-agent MDP with a time-varying reward $R_{i,h}(s, a \mid \pi_{[i-1]})$.

**Lemma 1** (Bellman Equation for Marginal Gains)**.** *For any policy $\pi_i$, $h \in [H]$, $s \in \mathcal{S}$, and $a \in \mathcal{A}$,*

$$V_h^{\pi_i}(s; \pi_{[i-1]}) = \sum_{a\in\mathcal{A}} \pi_{i,h}(s, a) Q_h^{\pi_i}(s, a; \pi_{[i-1]})$$

$$\begin{aligned} Q_h^{\pi_i}(s, a; \pi_{[i-1]}) &= R_{i,h}(s, a|\pi_{[i-1]}) \\ &+ \sum_{s'\in\mathcal{S}} P_{i,h}(s'|s, a) V_{h+1}^{\pi_i}(s'; \pi_{[i-1]}). \end{aligned}$$

*Moreover, let* $\widetilde{\pi}_i = \arg\max_{\pi_i} \mathbb{E}_{\pi_i}[\sum_{t=1}^{H} \Delta r_i(\mathbf{s}_t, \mathbf{a}_t)|s_1^i = \bar{s}_1^i, \pi_{[i-1]}]$. *Then*

$$V_h^{\widetilde{\pi}_i}(s; \pi_{[i-1]}) = \max_a Q_h^{\widetilde{\pi}_i}(s, a; \pi_{[i-1]})$$

$$Q_h^{\widetilde{\pi}_i}(s, a; \pi_{[i-1]}) = R_{i,h}(s, a|\pi_{[i-1]})$$
$$+ \sum_{s' \in \mathcal{S}} P_{i,h}(s'|s, a) V_{h+1}^{\widetilde{\pi}_i}(s'; \pi_{[i-1]}).$$

The proof sketch is deferred to Appendix B.3. Next, we discuss the algorithm of policy optimization in the case where the transition matrix $P$ is known.

### 4.2. Greedy Policy Optimization for known $P$

With a known transition dynamic $P$, we propose the algorithm `Greedy Policy Optimization` based on running policy optimization greedily, as described in Algorithm 1. The algorithm determines the policy of the agents in an iterative, greedy fashion. During the outer loop from Line 2 to Line 20, the algorithm greedily determines the policy for different agents sequentially by leveraging the submodular property of the reward function. For the inner loop from Line 3 to Line16, the algorithm finds the policy of agent $i$ by recursively optimizing the policy via backward induction from step $H$ to 1. After determine the policy of the $i$-th agent, the algorithm samples $N := \frac{1}{2\epsilon^2} \log \frac{2K|\mathcal{S}||\mathcal{A}|H}{\delta}$ trajectories of this agent from Line 17 to Line 19, which are used for estimating the marginal reward $R_{i,h}(s, a|\pi_{[i-1]})$ as in Line 8. This step is very important in achieving the computation complexity polynomial in $K$ since computing $R_{i,h}(s, a|\pi_{[i-1]})$ exactly requires marginalizing over all joint configurations of agents $1, \ldots, i$, incurring exponential complexity $O(|\mathcal{S}|^i|\mathcal{A}|^i)$. Instead, Line 8 uses sampling for estimation. This reduces computation complexity from exponential to polynomial: $O(K \cdot H \cdot |\mathcal{S}||\mathcal{A}| \cdot \max\{|\mathcal{S}|, \epsilon^{-2} \log(K|\mathcal{S}||\mathcal{A}|H/\delta)\})$, which is polynomial in all parameters. Below, we present the theoretical guarantees of the proposed algorithm.

**Theorem 1.** *Let us denote the optimal policy as $\pi^*$. When the number of samples $N$ in Algorithm 1 is set to $N = \frac{1}{2\epsilon^2} \log \frac{2K|\mathcal{S}||\mathcal{A}|H}{\delta}$, with probability at least $1 - \delta$, the policy $\pi$ produced by Algorithm 1 satisfies that*

$$V_1^{\pi}(\bar{\mathbf{s}}_1) \geq \frac{V_1^{\pi^*}(\bar{\mathbf{s}}_1)}{2} - \epsilon KH.$$

Theorem 1 guarantees a $1/2$-approximation with additive error $\epsilon KH$, matching the approximation ratio of greedy algorithms for monotone submodular maximization (Wolsey, 1982). The parameters $\epsilon$ and $\delta$ control the accuracy-complexity trade-off through the value of $N$. It is worth noting that although we restrict our policy to being decomposable, which indicates that the execution of individual agents

---

**Algorithm 1** `Greedy Policy Optimization`

1: $N := \frac{1}{2\epsilon^2} \log \frac{2K|\mathcal{S}||\mathcal{A}|H}{\delta}$
2: **for** $i = 1$ **to** $K$ **do**
3:   **for** $h = H$ **to** 1 **do**
4:     **for** $(s, a)$ **in** $\mathcal{S} \times \mathcal{A}$ **do**
5:       **if** $i = 1$ **then**
6:         $\widehat{R}_{1,h}(s, a) = f((s, a))$
7:       **else**
8:         Compute average marginal reward using sampled trajectories of first $i - 1$ agents: $\widehat{R}_{i,h}(s, a|\pi_{[i-1]}) = \frac{1}{N} \sum_{l=1}^{N} (f(\{(s_{h,l}^j, a_{h,l}^j)\}_{j \in [i-1]} \cup \{(s, a)\}) - f(\{(s_{h,l}^j, a_{h,l}^j)_{j \in [i-1]})\})$
9:       **end if**
10:      $\widehat{Q}_h^i(s, a) = \widehat{R}_{i,h}(s, a|\pi_{[i-1]}) + \sum_{s' \in \mathcal{S}} P_{i,h}(s'|s, a) \widehat{V}_{i,h+1}(s')$
11:     **end for**
12:     **for** $s$ **in** $\mathcal{S}$ **do**
13:       $\widehat{V}_h^i(s) = \max_{a \in \mathcal{A}} \widehat{Q}_h^i(s, a)$
14:       $\pi_{i,h}(s) = \arg\max_{a \in \mathcal{A}} \widehat{Q}_h^i(s, a)$
15:     **end for**
16:   **end for**
17:   **for** $l = 1$ **to** $N$ **do**
18:     Sample the $l$-th trajectory of agent $i$: $\tau_{i,l} = \{(s_{1,l}^i, a_{1,l}^i), (s_{2,l}^i, a_{2,l}^i), \ldots (s_{H,l}^i, a_{H,l}^i)\}$
19:   **end for**
20: **end for**
21: **return** policy $\pi$

---

doesn't depend on other agents, the output policy is competitive with respect to the optimal joint policy, which may be non-decomposable and generally requires exponential memory to represent and execute. This result demonstrates that decomposability does not inherently limit approximation quality in MARLS, and that near-optimal performance can be achieved within a tractable and practically implementable policy class.

### 4.3. UCB-Based Greedy Value Iteration for Unknown Transition Dynamics

In Section 4.2, we developed the `Greedy Policy Optimization` algorithm under the assumption of known transition dynamics. However, in many practical applications, the transition matrix $P$ is unknown and must be learned through interaction with the environment. In this setting, our objective shifts to minimizing regret, which is the cumulative difference between the optimal policy's performance and that of the learned policies during the exploration process.

To address this challenge, we propose `UCB-GVI`

(Upper Confidence Bound Greedy Value Iteration), a model-based reinforcement learning algorithm that combines optimistic exploration with greedy submodular maximization. The complete procedure is described in Algorithm 2.

**Algorithm description.** UCB-GVI operates over $T$ episodes, with each episode $k$ consisting of three phases: policy computation via optimistic value iteration, trajectory sampling for marginal reward estimation, and policy execution in the true environment.

First of all, the policy computation phase proceeds sequentially for each agent $i = 1, \ldots, K$ using backward value iteration from step $h = H$ to $h = 1$. For each state-action pair $(s, a)$ that has been visited in previous episodes (i.e., $N_{i,h}(s, a) > 0$), the algorithm first constructs an empirical transition model $\widehat{P}_{i,h}(s'|s, a)$ based on the observed transitions. The marginal reward $\widehat{R}_{i,h}(s, a|\pi_{[i-1]})$ is then estimated by sampling $N$ trajectories under the empirical dynamics $\widehat{P}_i$ and the previously computed policies $\pi_{[i-1]}$ of agents $1, \ldots, i - 1$ (Line 10). This sampling-based approach captures the expected marginal contribution of agent $i$ selecting action $a$ in state $s$, accounting for the stochastic behavior induced by both the transition dynamics and the policies of earlier agents in the sequential ordering.

Using these estimates, the algorithm computes optimistic Q-value estimates that incorporate exploration bonuses:

$$\widehat{Q}_{i,h}(s, a) = \widehat{R}_{i,h}(s, a|\pi_{[i-1]})$$
$$+ \sum_{s' \in \mathcal{S}} \widehat{P}_{i,h}(s'|s, a)\widehat{V}_{i,h+1}(s')$$
$$+ b(N_{i,h}(s, a)) + \frac{\epsilon}{KH},$$

where the exploration bonus $b(N) = H\sqrt{\frac{2S\iota}{N}} + \frac{3HS\iota}{N}$ with $\iota = \log(6S^2ATHK/\delta)$ provides an upper confidence bound on the estimation error. $N_{i,h}(s, a, s')$ denotes the number of times agent $i$ has transitioned from $(s, a)$ to $s'$ at step $h$ across all the previous episodes. For state-action pairs that have not been visited, the algorithm employs optimistic initialization $\widehat{Q}_h^i(s, a) = H$.

Next, the trajectory sampling phase generates synthetic data for marginal reward estimation in subsequent episodes. For each agent $i$, the algorithm samples $N = \frac{K^2H^2}{2\epsilon^2} \log \frac{6K|\mathcal{S}||\mathcal{A}|H}{\delta}$ simulated trajectories by executing policy $\pi_i$ under the empirical transition model $\widehat{P}_i$. Finally, in the policy execution phase from Line 26 to Line 33, the algorithm deploys the computed joint policy $\pi = (\pi_1, \ldots, \pi_K)$ in the true environment.

**Regret definition.** To formalize the algorithm's performance guarantee, we introduce the notion of $\alpha$-regret,

---

**Algorithm 2** Upper Confidence Bound Greedy Value Iteration (UCB-GVI)

1: **Input:** Episodes $T$, confidence $\delta$, approximation parameter $\epsilon$
2: **Initialize:** $\mathcal{D} \leftarrow \emptyset$, $N_{i,h}(s, a) \leftarrow 0$, $N_{i,h}(s, a, s') \leftarrow 0$ for all $i, h, s, a, s'$
3: Set $N \leftarrow \frac{K^2H^2}{2\epsilon^2} \log \frac{6K|\mathcal{S}||\mathcal{A}|H}{\delta}$
4: **for** episode $k = 1$ to $T$ **do**
5:   **for** agent $i = 1$ to $K$ **do**
6:     **for** step $h = H$ down to 1 **do**
7:       **for** $(s, a) \in \mathcal{S} \times \mathcal{A}$ **do**
8:         **if** $N_{i,h}(s, a) > 0$ **then**
9:           $\widehat{P}_{i,h}(s'|s, a) \leftarrow \frac{N_{i,h}(s,a,s')}{N_{i,h}(s,a)}$ for all $s' \in \mathcal{S}$
10:           $\widehat{R}_{i,h}(s, a|\pi_{[i-1]}) \leftarrow \frac{1}{N} \sum_{l=1}^{N} \Delta f(\{(\tilde{s}_{h,l}^m, \tilde{a}_{h,l}^m)\}_{m \in [i-1]}, (s, a))$
11:           $\widehat{Q}_{i,h}(s, a) \leftarrow \widehat{R}_{i,h}(s, a|\pi_{[i-1]}) + \sum_{s' \in \mathcal{S}} \widehat{P}_{i,h}(s'|s, a)\widehat{V}_{i,h+1}(s') + b(N_{i,h}(s, a)) + \frac{\epsilon}{KH}$
12:         **else**
13:           $\widehat{Q}_{i,h}(s, a) \leftarrow H$
14:         **end if**
15:       **end for**
16:       **for** $s \in \mathcal{S}$ **do**
17:         $\widehat{V}_{i,h}(s) \leftarrow \max_{a \in \mathcal{A}} \widehat{Q}_{i,h}(s, a)$
18:         $\pi_{i,h}(s) \leftarrow \arg\max_{a \in \mathcal{A}} \widehat{Q}_{i,h}(s, a)$
19:       **end for**
20:     **end for**
21:     **for** $l = 1$ to $N$ **do**
22:       Sample trajectory $\tau_{i,l} = \{(\tilde{s}_{h,l}^i, \tilde{a}_{h,l}^i)\}_{h=1}^{H}$ using $\pi_i$ and $\widehat{P}_i$
23:     **end for**
24:   **end for**
25:   Starting from initial state $\bar{s}_1^1, \ldots, \bar{s}_1^K$
26:   **for** step $h = 1$ to $H$ **do**
27:     **for** agent $i = 1$ to $K$ **do**
28:       Execute $a_{h,k}^i \leftarrow \pi_{i,h}(s_{h,k}^i)$; observe $s_{h+1,k}^i \sim P_{i,h}(\cdot|s_{h,k}^i, a_{h,k}^i)$
29:       $N_{i,h}(s_{h,k}^i, a_{h,k}^i) \leftarrow N_{i,h}(s_{h,k}^i, a_{h,k}^i) + 1$
30:       $N_{i,h}(s_{h,k}^i, a_{h,k}^i, s_{h+1,k}^i) \leftarrow N_{i,h}(s_{h,k}^i, a_{h,k}^i, s_{h+1,k}^i) + 1$
31:       $\mathcal{D} \leftarrow \mathcal{D} \cup \{(h, s_{h,k}^i, a_{h,k}^i, s_{h+1,k}^i)\}$
32:     **end for**
33:   **end for**
34: **end for**
35: **Return:** Policies $\{\pi^k\}_{k=1}^{T}$

which accounts for the approximation factor inherent in polynomial-time greedy algorithms for submodular maximization.

**Definition 3** ($\alpha$-regret). *Let $\pi^* = \arg\max_\pi V_1^\pi(\bar{\mathbf{s}}_1)$ denote the globally optimal policy and $\pi^k$ denote the policy executed in episode $k$. The $\alpha$-regret of an algorithm over $T$ episodes is*

$$\mathcal{R}_{T,\alpha} = \sum_{k=1}^{T} \left[ \alpha V_1^{\pi^*}(\bar{\mathbf{s}}_1) - V_1^{\pi^k}(\bar{\mathbf{s}}_1) \right].$$

The parameter $\alpha \in (0, 1]$ characterizes the approximation guarantee: $\alpha = 1$ recovers the standard regret notion, while $\alpha < 1$ reflects computational constraints. For our problem, the greedy algorithm achieves a $1/2$-approximation to the optimal policy under the empirical model. In particular, our main theoretical contribution establishes a near-optimal regret bound for `UCB-GVI`. Before we present the results in Theorem 2, below we illustrate the technical novelty of our proof.

**Technical Overview** The proof of our regret bound (Appendix D) requires fundamentally new techniques beyond standard UCB-type analyses for single-agent reinforcement learning (Azar et al., 2017; Jin et al., 2018). While our approach builds on optimistic value estimation, the multi-agent and submodular structure of MARLS introduces several fundamental challenges that necessitate new analytical tools. In particular, our analysis is different from existing work along three dimensions: (i) establishing approximation guarantees under empirical transition dynamics, (ii) controlling multi-agent transition estimation errors without incurring exponential dependence on the number of agents $K$ and (iii) estimating marginal rewards through an additional sampling procedure within the learning loop.

Our analysis consists of two main components.

**Part I: Approximation Ratio under Empirical Dynamics**. We prove that the greedy sequential policy construction achieves a $\frac{1}{2}$-approximation on the auxiliary MDP $\widehat{M}$ with transition dynamics $\widehat{\mathbf{P}}^k$ and augmented reward $r(\mathbf{s}, \mathbf{a}) + \sum_{i=1}^{K} b(N_{i,h}^k(s^i, a^i))$ (Lemma 8). The central challenge is that the optimal policy $\pi^*$ may be **non-decomposable**, which means that its value function cannot be expressed as a sum of marginal gains from individual agents. To overcome this, we introduce a novel proof technique comparing two hypothetical agent groups: one following the greedy policy $\pi^k$ and another following the optimal policy $\pi^*$. We carefully define the quantity $G_i^k$ which captures the marginal contribution of adding agent $i$ from the $\pi^*$ group (see the definition in (14)). By exploiting submodularity and monotonicity, we establish $\sum_{i=1}^{K} G_i^k \geq \bar{V}_1^{\pi^*,k}(\bar{\mathbf{s}}_1) - \bar{V}_1^{\pi^k,k}(\bar{\mathbf{s}}_1)$ while simultaneously

bounding each $G_i^k \leq \bar{V}_1^{\pi_i^k,k}(\bar{s}_1^i; \pi_{[i-1]}^k) + O(\epsilon/K)$. Telescoping over agents then yields the desired approximation ratio.

**Part II: Regret Decomposition via Multi-Agent Telescoping and Error Isolation**. We obtain the final regret bound by recursively expanding the value gap $\bar{V}_h^{\pi^k,k}(\mathbf{s}_h^k) - V_h^{\pi^k}(\mathbf{s}_h^k)$ backward from $h = H$ to $h = 1$. The critical technical challenge is bounding the difference between expected values under empirical versus true transitions for non-negative functions as given in Lemma 10 and Lemma 9. Both bounds exploit the product structure $\mathbf{P}_h = \prod_{i=1}^{K} P_{i,h}$ via telescoping decomposition, isolating individual agent errors to avoid exponential dependence on $K$. Lemma 9 employs a more complex and refined analysis: it first establishes a coarse recursive relation $Y_i - Y_{i-1} \leq \frac{Y_{i-1}}{2HK} + O(\frac{\iota}{N})$ through calibrated concentration inequalities, then refines to $Y_i \leq (1 + \frac{1}{K})Y_{i-1} + O(\frac{\iota}{N})$ for the tight bound. The interplay between these two bounds—using Lemma 10 to control deterministic exploration costs and Lemma 9 to enable self-normalized recursive analysis—ultimately yields the $\tilde{O}(S^2 A H^3 K^2 \sqrt{T})$ regret rate.

The theoretical guarantee of `Greedy Policy Optimization` is provided in Theorem 2.

**Theorem 2** (Regret bound for `UCB-GVI`). *With probability at least $1 - \delta$, Algorithm 2 achieves a regret bound of*

$$\mathcal{R}_{T,1/2} = O\Big( S^2 A H^3 K^2 \log(SATHK/\delta) \log T$$
$$+ H^2 K S \sqrt{AT \log(SATHK/\delta)} \Big),$$

*where $O(\cdot)$ suppresses poly-logarithmic factors.*

The proof of the theorem is deferred to Appendix D. From the result, we have that when $K = 1$, our bound reduces to $O(H^2 S \sqrt{AT})$ up to logarithmic factors, which recovers the single-agent setting. Compared to the lower bound and the best known upper bounds of order $\Omega(H\sqrt{SAHT})$ for standard episodic reinforcement learning (Domingues et al., 2021), (Azar et al., 2017), our bound exhibits an additional $\sqrt{HS}$ factor, reflecting the challenges of multi-agent coordination and marginal reward estimation.

Importantly, the regret scales only polynomially with the number of agents $K$, demonstrating that meaningful learning is possible in this setting despite the exponential size of the joint action space. In particular, the dominant term $O(H^2 K S \sqrt{AT})$ scales linearly with $K$, indicating that the learning cost per agent is comparable to that of independent single-agent learning, even though agents interact through a shared submodular reward. The additional logarithmic term with quadratic dependence on $K$ further reflects the computational-approximation tradeoff inherent in polynomial-time algorithms for submodular maximization under matroid constraints.

## Acknowledgements

Victoria Crawford is supported in part by the Texas A&M University Division of Research Targeted Proposal Teams (TPT) funding program, and the Seed Program for AI, Computing, and Data Science created by the Texas A&M Institute for Data Science. Yi Zhou's work is supported by the National Science Foundation under Grants DMS-2134223 and ECCS-2237830.

## Impact Statement

This paper presents work whose goal is to advance the field of machine learning. There are many potential societal consequences of our work, none of which we feel must be specifically highlighted here.

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

# Appendix

## A. Additional Related Work

### A.1. Combinatorial Multi-Armed Bandit Problem with Submodular Reward

Submodular rewards have been introduced in the multi-armed combinatorial bandit setting (Chen et al., 2025b; Takemori et al., 2020; Chen and Crawford, 2024; Singla et al., 2016). There are two primary lines of research in this area. The first focuses on the online regret-minimization setting, where the objective is to interact sequentially with an unknown environment and minimize cumulative regret (Takemori et al., 2020; Fourati et al., 2023; 2024; Yue and Guestrin, 2011). The second line of work studies the pure-exploration setting, where the objective is to find a high-quality solution in as few noisy samples as possible(Singla et al., 2016; Chen and Crawford, 2024). While these problem settings are related, they differ fundamentally in problem complexity: bandit settings lack Markovian state dynamics and consider only single-step episodes in which agents select subsets of arms and immediately observe stochastic rewards. In contrast, our framework incorporates full MDP dynamics with state transitions, multi-step planning horizons of length $H$, and coordination among multiple agents whose joint actions influence both immediate rewards and future state distributions.

### A.2. Regret Analysis in Tabular RL

The study of regret minimization in episodic Markov decision processes (MDPs) has a long and rich history in reinforcement learning theory (Osband et al., 2016; Jin et al., 2018; Zhang et al., 2021; Dann et al., 2021; Agrawal and Agrawal, 2025; Neu and Pike-Burke, 2020). For episodic MDPs with horizon $H$, $S$ states, $A$ actions, and $T$ total interaction steps, Azar et al. (2017) established a near-minimax regret lower bound of $\Omega(\sqrt{H^2 SAT})$ and proposed the model-based algorithm UCBVI, which achieves a matching upper bound up to logarithmic factors. Zanette and Brunskill (2019) proposed the EULER algorithm, which matches the minimax rate and yields tighter instance-dependent bounds by exploiting Bernstein-type concentration and the variance structure of the value function. Complementary to these model-based approaches, Jin et al. (2018) developed the first provably efficient model-free algorithm via Q-learning with UCB exploration bonuses, achieving $\tilde{O}(\sqrt{H^3 SAT})$ regret. Subsequent work reduced the horizon dependence through refined exploration strategies (Zhang et al., 2020) and obtained a bound that matches the near-minimax lower bound.

## B. Appendix for Section 4.1

In this portion of the appendix, we present missing details and proofs from Section 4.1 in the main paper. We first present missing content from Section 4.1.1 in Section B.1 and B.2, then provide the omitted proof for Lemma 1 in Section B.3.

### B.1. Bellman Optimality

**Lemma 2** (Bellman Optimality (Bellman, 1957; Sutton et al., 1998)). *For any policy $\pi$, we have that*

$$V_h^\pi(\mathbf{s}) = \sum_{a^1, a^2, \ldots, a^K \in \mathcal{A}} \pi_h(\mathbf{s}, \mathbf{a}) Q_h^\pi(\mathbf{s}, \mathbf{a}),$$

$$Q_h^\pi(\mathbf{s}, \mathbf{a}) = r(\mathbf{s}, \mathbf{a}) + \sum_{s'^1, s'^2, \ldots, s'^K \in \mathcal{S}} \prod_{i=1}^{K} P_{i,h}(s'^i | s^i, a^i) V_{h+1}^\pi(\mathbf{s}', \mathbf{a}). \tag{3}$$

*Besides, there exists a deterministic optimal policy $\pi^*$ such that $\pi_h^*(\mathbf{s}, \mathbf{a}) = 1$ or $0$. In particular, We denote the optimal value function and Q function as $V^*$ and $Q^*$. Then the optimal solution satisfies*

$$V_h^*(\mathbf{s}) = \max_{\mathbf{a} \in \mathcal{A}^K} Q_h^*(\mathbf{s}, \mathbf{a}),$$

$$Q_h^*(\mathbf{s}, \mathbf{a}) = r(\mathbf{s}, \mathbf{a}) + \sum_{s'^1, s'^2, \ldots, s'^K \in \mathcal{S}} \prod_{i=1}^{K} P_{i,h}(s'^i | s^i, a^i) V_{h+1}^*(\mathbf{s}'). \tag{4}$$

**B.2. Special case of $H = 1$**

In this portion of the appendix, we prove that when $H = 1$, the problem of finding optimal policies in MARLS is NP-hard by reduction to submodular maximization under partition matroid constraints. In particular, we have the following lemma.

**Lemma 3** (Equivalence to Partition Matroid Submodular Maximization). *Let $f : 2^{S \times A} \to [0, 1]$ be monotone and submodular. The problem of finding the optimal policy when $H = 1$,*

$$\max_{a^1, \ldots, a^K} f\big(\{(\bar{s}^1, a^1), \ldots, (\bar{s}^K, a^K)\}\big),$$

*is equivalent to monotone submodular maximization subject to a partition matroid constraint.*

*Proof.* We establish the equivalence by constructing a partition matroid and a corresponding monotone submodular function. For each agent $i \in [K]$, define $\mathcal{A}_i = \{(i, a) : a \in \mathcal{A}\}$ as a labeled copy of the action space. Let the ground set be $\mathcal{U} = \bigcup_{i=1}^K \mathcal{A}_i$, then $\{A_i\}_{i \in [K]}$ form a partition of $\mathcal{U}$, as $A_i$ and $A_j$ are disjoint for any $i \neq j$. We define the partition matroid $\mathcal{M} = (\mathcal{U}, \mathcal{I})$ with independent sets

$$\mathcal{I} = \{X \subseteq \mathcal{U} : |X \cap \mathcal{A}_i| \leq 1 \text{ for all } i \in [K]\}.$$

This constraint ensures that at most one action from each agent's action space can be selected.

For any $X \subseteq \mathcal{U}$, let $X_i = X \cap \mathcal{A}_i$ denote the elements in $X$ corresponding to agent $i$. Define $F : 2^{\mathcal{U}} \to [0, 1]$ by

$$F(X) = f\big(\{(\bar{s}^i, a) : (i, a) \in X\}\big).$$

If $X_i = \emptyset$, agent $i$ contributes no state-action pair to the argument of $f$.

We first verify that $F$ is monotone submodular. For monotonicity, if $X \subseteq Y \subseteq \mathcal{U}$, then $\{(\bar{s}^i, a) : (i, a) \in X\} \subseteq \{(\bar{s}^i, a) : (i, a) \in Y\}$, so $F(X) \leq F(Y)$ by monotonicity of $f$. For submodularity, consider $X \subseteq Y \subseteq \mathcal{U}$ and $e = (j, a) \in \mathcal{U} \setminus Y$. Define $S_X = \{(\bar{s}^i, a') : (i, a') \in X\}$ and $S_Y = \{(\bar{s}^i, a') : (i, a') \in Y\}$. Then $S_X \subseteq S_Y$ and $(\bar{s}^j, a) \notin S_Y$. By submodularity of $f$,

$$F(X \cup \{e\}) - F(X) = f(S_X \cup \{(\bar{s}^j, a)\}) - f(S_X)$$
$$\geq f(S_Y \cup \{(\bar{s}^j, a)\}) - f(S_Y) = F(Y \cup \{e\}) - F(Y).$$

Next, we prove the equivalence. Since $F$ is monotone, there exists at least one optimal solution $X^*$ of the problem $\max_{X \in \mathcal{I}} F(X)$ with the form $X^* = \{(1, a^1), \ldots, (K, a^K)\}$, i.e., containing exactly one element from each partition $\mathcal{A}_i$. Therefore,

$$\max_{X \in \mathcal{I}} F(X) = \max_{X \subseteq \mathcal{U} : |X \cap \mathcal{A}_i| = 1 \text{ for all } i} F(X).$$

Since the optimization problem $\max_{a^1, \ldots, a^K \in \mathcal{A}} f\big(\{(\bar{s}^i, a^i) : i \in [K]\}\big)$ is equivalent to $\max_{X \subseteq \mathcal{U} : |X \cap \mathcal{A}_i| = 1 \text{ for all } i} F(X)$, we conclude that

$$\max_{a^1, \ldots, a^K \in \mathcal{A}} f\big(\{(\bar{s}^i, a^i) : i \in [K]\}\big) = \max_{X \in \mathcal{I}} F(X),$$

which is monotone submodular maximization subject to a partition matroid constraint. $\square$

**B.3. Proof of Lemma 1**

In this section, we prove the omitted proof sketch of Lemma 1.

*Proof sketch.* Given fixed $\pi_{[i-1]}$, for the first part, by definition of $Q_h^{\pi_i}$ and the tower property of expectation,

$$Q_h^{\pi_i}(s, a; \pi_{[i-1]}) = \mathbb{E}_{\pi_i}[\Delta r_i(\mathbf{s}_h, \mathbf{a}_h) + \sum_{t=h+1}^H \Delta r_i(\mathbf{s}_t, \mathbf{a}_t) \mid s_h^i = s, a_h^i = a, \pi_{[i-1]}]$$
$$= R_{i,h}(s, a | \pi_{[i-1]}) + \mathbb{E}_{s_{h+1}^i \sim P_{i,h}(\cdot|s,a)}[V_{h+1}^{\pi_i}(s_{h+1}^i; \pi_{[i-1]})]$$
$$= R_{i,h}(s, a | \pi_{[i-1]}) + \sum_{s' \in \mathcal{S}} P_{i,h}(s'|s, a) V_{h+1}^{\pi_i}(s'; \pi_{[i-1]}).$$

The equation for $V_h^{\pi_i}$ follows from the law of total expectation.

Let us define an augmented MDP $M_i$ for agent $i$ with state space $\mathcal{S}$, action space $\mathcal{A}$, transition kernel $P_{i,h}$, and time-varying reward $R_{i,h}(\cdot, \cdot | \pi_{[i-1]})$. The key observation is that $R_{i,h}(s, a | \pi_{[i-1]})$ is deterministic (as a function of $s, a, h$) because the expectation in (2) is taken over the fixed stochastic policy $\pi_{[i-1]}$ and known transition dynamics. It then follows from the result above that $Q_h^{\pi_i}(s, a; \pi_{[i-1]})$ and $V_h^{\pi_i}(s; \pi_{[i-1]})$ are the value functions for $M_i$ given policy $\pi_i$. By definition of $\widetilde{\pi}_i$, we know that $\widetilde{\pi}_i$ is the optimal policy for MDP $M_i$. By the Bellman optimality principle, the optimal value function satisfies $V_h^{\widetilde{\pi}_i}(s; \pi_{[i-1]}) = \max_a Q_h^{\widetilde{\pi}_i}(s, a; \pi_{[i-1]})$, and the optimal Q-function satisfies the Bellman equation. This can be verified by backward induction from $h = H$ to $h = 1$. $\qquad\square$

## C. Appendix for Section 4.2

In this section, we present the proof of the main result in Theorem 1.

First of all, we present the following lemma that guarantees the high probability event during the execution of `Greedy Policy Optimization`.

**Lemma 4.** *Let us define the event*

$$\mathcal{E} := \left\{ \forall (s, a) \in \mathcal{S} \times \mathcal{A}, \forall i \in [K], h \in [H] : |R_{i,h}(s, a | \pi_{[i-1]}) - \widehat{R}_{i,h}(s, a | \pi_{[i-1]})| \leq \epsilon \right\}.$$

*Then $\mathcal{E}$ holds with probability $1 - \delta$.*

*Proof.* For any $i \in [K]$, $h \in [H]$, and $l \in [N]$, we consider the following random variable obtained from the $l$-th trajectory $\{(s_{h,l}^j, a_{h,l}^j)\}_{h \in [H], j \in [K]}$.

$$f(\{(s_{h,l}^j, a_{h,l}^j)\}_{j \in [i-1]} \cup \{(s, a)\}) - f(\{(s_{h,l}^j, a_{h,l}^j)\}_{j \in [i-1]}) = \Delta f(\{(s_{h,l}^j, a_{h,l}^j)\}_{j \in [i-1]}, (s, a))$$

Then it follows that

$$\mathbb{E}[f(\{(s_{h,l}^j, a_{h,l}^j)\}_{j \in [i-1]} \cup \{(s, a)\}) - f(\{(s_{h,l}^j, a_{h,l}^j)\}_{j \in [i-1]})] = \mathbb{E}[\Delta f(\{(s_{h,l}^j, a_{h,l}^j)\}_{j \in [i-1]}, (s, a))]$$
$$= R_{i,h}(s, a | \pi_{[i-1]})$$

By Hoeffding's inequality, we have that for any fixed $(s, a) \in \mathcal{S} \times \mathcal{A}$, and $h \in [H]$, the following inequality holds with probability at least $1 - \frac{\delta}{K|\mathcal{S}||\mathcal{A}|H}$:

$$|R_{i,h}(s, a | \pi_{[i-1]}) - \widehat{R}_{i,h}(s, a | \pi_{[i-1]})| \leq \epsilon$$

By taking the union bound over all $i \in [K]$, $h \in [H]$, $(s, a) \in \mathcal{S} \times \mathcal{A}$, we have that the event $\mathcal{E}$ holds with probability at least $1 - \delta$. $\qquad\square$

Next, we provide the major proofs of Theorem 1.

*Proof.* Suppose we have two groups of $K$ agents, the first group of $K$ agents follows the policy output by Algorithm 1. We denote the trajectory by $\{(s_h^i, a_h^i)\}_{h \in [H], i \in [K]}$. The second group of $K$ agents follows the optimal policy $\pi^*$, and we denote the trajectory of the second group of agents by $\{(\tilde{s}_h^i, \tilde{a}_h^i)\}_{h \in [H], i \in [K]}$. For notation simplicity, we define

$$G_i = \mathbb{E}[\sum_{h=1}^H \Delta f(\{(s_h^j, a_h^j)\}_{j \in [K]} \cup \{(\tilde{s}_h^m, \tilde{a}_h^m)\}_{m \in [i-1]}, (\tilde{s}_h^i, \tilde{a}_h^i)) | (\mathbf{s}, \mathbf{a}) \sim \pi, (\tilde{\mathbf{s}}, \tilde{\mathbf{a}}) \sim \pi^*],$$

where $(\mathbf{s}, \mathbf{a}) \sim \pi, (\tilde{\mathbf{s}}, \tilde{\mathbf{a}}) \sim \pi^*$ means that $\{(s_h^i, a_h^i)\}_{h \in [H], i \in [K]}$ is sampled according to policy $\pi$, and $\{(\tilde{s}_h^i, \tilde{a}_h^i)\}_{h \in [H], i \in [K]}$ is sampled based on optimal policy $\pi^*$. Then by definition, we have that

$$G_i = \mathbb{E}[\sum_{h=1}^H f(\{(s_h^j, a_h^j)\}_{j \in [K]} \cup \{(\tilde{s}_h^m, \tilde{a}_h^m)\}_{m \in [i]}) - f(\{(s_h^j, a_h^j)\}_{j \in [K]} \cup \{(\tilde{s}_h^m, \tilde{a}_h^m)\}_{m \in [i-1]}) | (\mathbf{s}, \mathbf{a}) \sim \pi, (\tilde{\mathbf{s}}, \tilde{\mathbf{a}}) \sim \pi^*].$$

It then follows that

$$
\begin{aligned}
\sum_{i=1}^{K} G_i &= \mathbb{E}[\sum_{h=1}^{H} f(\{(s_h^j, a_h^j)\}_{j \in [K]} \cup \{(\tilde{s}_h^m, \tilde{a}_h^m)\}_{m \in [K]}) - f(\{(s_h^j, a_h^j)\}_{j \in [K]}) | (\mathbf{s}, \mathbf{a}) \sim \pi, (\tilde{\mathbf{s}}, \tilde{\mathbf{a}}) \sim \pi^*] \\
&\geq \mathbb{E}[\sum_{h=1}^{H} f(\{(\tilde{s}_h^m, \tilde{a}_h^m)\}_{m \in [K]}) - f(\{(s_h^j, a_h^j)\}_{j \in [K]}) | (\mathbf{s}, \mathbf{a}) \sim \pi, (\tilde{\mathbf{s}}, \tilde{\mathbf{a}}) \sim \pi^*] \\
&= \mathbb{E}[\sum_{h=1}^{H} f(\{(\tilde{s}_h^m, \tilde{a}_h^m)\}_{m \in [K]}) | (\tilde{\mathbf{s}}, \tilde{\mathbf{a}}) \sim \pi^*] - \mathbb{E}[\sum_{h=1}^{H} f(\{(s_h^j, a_h^j)\}_{j \in [K]}) | (\mathbf{s}, \mathbf{a}) \sim \pi] \\
&= V_1^*(\bar{\mathbf{s}}_1) - V_1^{\pi}(\bar{\mathbf{s}}_1),
\end{aligned}
\tag{5}
$$

where the inequality follows from the fact that $f$ is monotone. Next, we bound $G_i$ for each $i$. Since $\{(s_h^j, a_h^j)\}_{j \in [i-1]} \subseteq \{(s_h^j, a_h^j)\}_{j \in [K]} \cup \{(\tilde{s}_h^m, \tilde{a}_h^m)\}_{m \in [i]}$, by submodularity, we have that

$$
G_i \leq \mathbb{E}[\sum_{h=1}^{H} \Delta f(\{(s_h^j, a_h^j)\}_{j \in [i-1]}, (\tilde{s}_h^i, \tilde{a}_h^i)) | (\mathbf{s}, \mathbf{a}) \sim \pi, (\tilde{\mathbf{s}}, \tilde{\mathbf{a}}) \sim \pi^*].
\tag{6}
$$

Notice that the optimal policy is not guaranteed to be decomposable. Let us denote the policy $\pi_{i,h}^*$ to be the marginal policy of the $i$-th agent at time step $h$ while the global policy is $\pi^*$. Then the marginal policy of the agent $i$ should be computed via integration of the state-action pairs on the other agents. In particular,

$$
\pi_{i,h}^*(s, a) = \sum_{\substack{\mathbf{s}, \mathbf{a} \\ s^i = s, a^i = a}} \pi_h^*(\mathbf{s}, \mathbf{a}).
$$

For notational simplicity, here we also use $\pi_i^* = \{\pi_{i,h}^*\}_{h \in [H]}$ to denote the marginal policy of agent $i$ throughout the entire episode. Therefore,

$$
\begin{aligned}
&\mathbb{E}[\sum_{h=1}^{H} \Delta f(\{(s_h^j, a_h^j)\}_{j \in [i-1]}, (\tilde{s}_h^i, \tilde{a}_h^i)) | (\mathbf{s}, \mathbf{a}) \sim \pi, (\tilde{\mathbf{s}}, \tilde{\mathbf{a}}) \sim \pi^*] \\
=&\mathbb{E}[\sum_{h=1}^{H} \Delta f(\{(s_h^j, a_h^j)\}_{j \in [i-1]}, (\tilde{s}_h^i, \tilde{a}_h^i)) | (\mathbf{s}, \mathbf{a}) \sim \pi, (\tilde{s}^i, \tilde{a}^i) \sim \pi_i^*] \\
=&V_1^{\pi_i^*}(\bar{s}_1; \pi_{[i-1]}).
\end{aligned}
$$

Combine the above inequality with (6), it then follows that

$$
G_i \leq V_1^{\pi_i^*}(\bar{s}_1; \pi_{[i-1]}).
\tag{7}
$$

Next, we introduce and explain the following Lemma 5. The proof of the lemma is deferred later in this section.

**Lemma 5.** *With probability at least $1 - \delta$, we have that for any time step $h \in [H]$, and any agent $i \in [K]$,*

$$
V_h^{\pi_i^*}(s; \pi_{[i-1]}) \leq \widehat{V}_h^i(s) + \epsilon(H - h + 1) \leq V_h^{\pi_i}(s; \pi_{[i-1]}) + 2\epsilon(H - h + 1)
$$
$$
Q_h^{\pi_i^*}(s, a; \pi_{[i-1]}) \leq \widehat{Q}_h^i(s, a) + \epsilon(H - h + 1) \leq Q_h^{\pi_i}(s, a; \pi_{[i-1]}) + 2\epsilon(H - h + 1).
$$

This lemma implies that the estimate of value function $\widehat{V}_h^i(s)$ and $\widehat{Q}_h^i(s, a)$ indeed lie in between the true value function of the optimal marginal policy $\pi_i^*$ and $\pi_i$, and that the output policy $\pi_i$ is approximately better than $\pi_i^*$. Combine the result in the lemma with (7), we would have that

$$
G_i \leq V_1^{\pi_i}(\bar{s}_1^i; \pi_{[i-1]}) + 2\epsilon H
$$

Since $\sum_{i=1}^{K} G_i \geq V_1^*(\bar{s}_1) - V_1^\pi(\bar{s}_1)$ from (5), we have that

$$V_1^*(\bar{s}_1) - V_1^\pi(\bar{s}_1) \leq \sum_{i=1}^{K}\{V_1^{\pi_i}(\bar{s}_1^i; \pi_{[i-1]}) + 2\epsilon H\}$$
$$\leq V_1^\pi(\bar{s}_1) + 2\epsilon KH.$$

By rearranging the inequality above, we can prove the result in the theorem.

$\square$

Next, we present the omitted proof of Lemma 5, which is restated as follows.

**Lemma 5.** *With probability at least $1 - \delta$, we have that for any time step $h \in [H]$, and any agent $i \in [K]$,*

$$V_h^{\pi_i^*}(s; \pi_{[i-1]}) \leq \widehat{V}_h^i(s) + \epsilon(H - h + 1) \leq V_h^{\pi_i}(s; \pi_{[i-1]}) + 2\epsilon(H - h + 1)$$
$$Q_h^{\pi_i^*}(s, a; \pi_{[i-1]}) \leq \widehat{Q}_h^i(s, a) + \epsilon(H - h + 1) \leq Q_h^{\pi_i}(s, a; \pi_{[i-1]}) + 2\epsilon(H - h + 1)$$

*Proof.* We prove the lemma by induction. First of all, when $h = H$, by definition, we have that

$$Q_H^{\pi_i^*}(s, a; \pi_{[i-1]}) = Q_H^{\pi_i}(s, a; \pi_{[i-1]}) = R_{i,H}(s, a | \pi_{[i-1]}),$$

and

$$\widehat{Q}_H^i(s, a) = \widehat{R}_{i,H}(s, a | \pi_{[i-1]})$$

From Lemma 4, we have that with probability at least $1 - \delta$, it holds that

$$Q_H^{\pi_i^*}(s, a; \pi_{[i-1]}) \leq \widehat{Q}_H^i(s, a) + \epsilon \leq Q_H^{\pi_i}(s, a; \pi_{[i-1]}) + 2\epsilon. \tag{8}$$

By the Bellman's equation in Lemma 1, we have that $V_H^{\pi_i^*}(s; \pi_{[i-1]}) = \sum_{a \in \mathcal{A}} \pi_{i,H}^*(s, a) Q_H^{\pi_i^*}(s, a; \pi_{[i-1]})$. Since $\sum_{a \in \mathcal{A}} \pi_{i,h}^*(s, a) = 1$, we have

$$V_H^{\pi_i^*}(s; \pi_{[i-1]}) \leq \max_{a \in \mathcal{A}} Q_H^{\pi_i^*}(s, a; \pi_{[i-1]})$$
$$\leq \max_{a \in \mathcal{A}} \widehat{Q}_H^i(s, a) + \epsilon$$
$$= \widehat{V}_H^i(s) + \epsilon,$$

where the second inequality follows from the inequality in (8), and the last equation follows from the definition of $\widehat{V}_H^i(s)$. By definition of the output policy $\pi$, we have that $\widehat{V}_H^i(s) = \widehat{Q}_H^i(s, \pi_{i,H}(s))$, and $V_H^{\pi_i}(s; \pi_{[i-1]}) = Q_H^{\pi_i}(s, \pi_{i,H}(s); \pi_{[i-1]})$. It then follows that

$$\widehat{V}_H^i(s) = \widehat{Q}_H^i(s, \pi_{i,h}(s))$$
$$\leq Q_H^{\pi_i}(s, \pi_{i,h}(s); \pi_{[i-1]}) + \epsilon$$
$$= V_H^{\pi_i}(s; \pi_{[i-1]}) + \epsilon$$

where the last inequality follows from Bellman's equation in Lemma 1. From the result in (8), we have that the lemma holds for $h = H$.

Suppose the lemma holds for the case of the step $h + 1$. Then for time step $h$. By the Bellman's equation in Lemma 1, we

have that

$$Q_h^{\pi_i^*}(s,a;\pi_{[i-1]}) = R_{i,h}(s,a|\pi_{[i-1]}) + \sum_{s' \in \mathcal{S}} P_{i,h}(s'|s,a) V_{h+1}^{\pi_i^*}(s';\pi_{[i-1]})$$

$$\leq R_{i,h}(s,a|\pi_{[i-1]}) + \sum_{s' \in \mathcal{S}} P_{i,h}(s'|s,a)\{\widehat{V}_{h+1}^i(s') + \epsilon(H-h)\}$$

$$\leq \widehat{R}_{i,h}(s,a|\pi_{[i-1]}) + \epsilon + \sum_{s' \in \mathcal{S}} P_{i,h}(s'|s,a)\{\widehat{V}_{h+1}^i(s') + \epsilon(H-h)\}$$

$$\leq \widehat{Q}_h^i(s,a) + \epsilon(H-h+1)$$

where the first equation follows from the Bellman's equation in Lemma 1, with the policy of the $i$-th agent being $\pi_i^*$. The second inequality follows from the event $\mathcal{E}$, and the last inequality follows from the fact that $\sum_{s' \in \mathcal{S}} P_{i,h}(s'|s,a) = 1$ for any $(s,a) \in \mathcal{S} \times \mathcal{A}$, $h \in [H]$ and $i \in [K]$. Besides, from the algorithm description in Line 10 in Algorithm 1, we have that

$$\widehat{Q}_h^i(s,a) = \widehat{R}_{i,h}(s,a|\pi_{[i-1]}) + \sum_{s' \in \mathcal{S}} P_{i,h}(s'|s,a)\widehat{V}_{h+1}^i(s').$$

Since by Lemma 1 we have that $Q_h^{\pi_i}(s,a;\pi_{[i-1]}) = R_{i,h}(s,a|\pi_{[i-1]}) + \sum_{s' \in \mathcal{S}} P_{i,h}(s'|s,a) V_{h+1}^{\pi_i}(s';\pi_{[i-1]})$. By the assumption that the results holds for $h+1$, we have that

$$V_{h+1}^{\pi_i^*}(s;\pi_{[i-1]}) \leq \widehat{V}_{h+1}^i(s) + \epsilon(H-h) \leq V_{h+1}^{\pi_i}(s;\pi_{[i-1]}) + 2\epsilon(H-h).$$

Then

$$\widehat{Q}_h^i(s,a) = \widehat{R}_{i,h}(s,a|\pi_{[i-1]}) + \sum_{s' \in \mathcal{S}} P_{i,h}(s'|s,a)\widehat{V}_{h+1}^i(s')$$

$$\leq R_{i,h}(s,a|\pi_{[i-1]}) + \sum_{s' \in \mathcal{S}} P_{i,h}(s'|s,a)\{V_{h+1}^{\pi_i}(s';\pi_{[i-1]}) + \epsilon(H-h)\} + \epsilon$$

$$\leq Q_h^{\pi_i}(s,a;\pi_{[i-1]}) + \epsilon(H-h+1).$$

Next, by Lemma 1, we have

$$V_h^{\pi_i^*}(s;\pi_{[i-1]}) = \sum_{a \in \mathcal{A}} \pi_{i,h}^*(s,a) Q_h^{\pi_i^*}(s,a;\pi_{[i-1]})$$

$$\leq \max_{a \in \mathcal{A}} Q_h^{\pi_i^*}(s,a;\pi_{[i-1]})$$

$$\leq \max_{a \in \mathcal{A}} \widehat{Q}_h^i(s,a) + \epsilon(H-h+1)$$

$$= \widehat{V}_h^i(s,a) + \epsilon(H-h+1).$$

Since

$$\widehat{V}_h^i(s) = \widehat{Q}_h^i(s,\pi_{i,h}(s))$$

$$\leq Q_h^{\pi_i}(s,\pi_{i,h}(s);\pi_{[i-1]}) + \epsilon(H-h+1)$$

$$= V_h^{\pi_i}(s;\pi_{[i-1]}) + \epsilon(H-h+1)$$

Thus, we can prove the lemma. □

# D. Appendix for Section 4.3

In this section, we present the omitted proof and discussion for Section 4.3. In particular, we present the omitted proof of Theorem 2. First of all, we define and prove the high probability events.

## D.1. High Probability Event

Before we prove the theorem, we introduce several definitions. First of all, let us define the policy executed in time step $h$ for the $k$-th episode for agent $i$ as $\pi_{i,h}^k$. Then we denote the full policy of agent $i$ at episode $k$ as $\pi_i^k := \{\pi_{i,h}^k\}_{h\in[H]}$. Furthermore, we denote the empirical estimate of the transition matrix as $\widehat{P}_{i,h}^k$. Let $\widehat{R}_{i,h}^k(s,a|\pi_{[i-1]}^k)$ denote the estimated marginal reward computed at Line 10 in episode $k$ of Algorithm 2. This estimate is obtained by sampling trajectories from policy $\pi^k$ using the empirical transition model $\widehat{P}^k$. Define $R_{i,h}^k(s,a|\pi_{[i-1]}^k) = \mathbb{E}_{\widehat{P}^k}[\Delta r_i(\mathbf{s}_h,\mathbf{a}_h)|s_h^i = s, a_h^i = a, \pi_{[i-1]}^k]$ as the expected marginal reward under $\widehat{P}^k$ and $\pi_{[i-1]}^k$. By construction, $\mathbb{E}[\widehat{R}_{i,h}^k(s,a|\pi_{[i-1]}^k)] = R_{i,h}^k(s,a|\pi_{[i-1]}^k)$, where the expectation is over the trajectory samples used to compute $\widehat{R}_{i,h}^k$.

We define the *high probability event* $\mathcal{E} := \mathcal{E}_1 \cap \mathcal{E}_2$ with $\iota := \log(6S^2ATHK/\delta)$:

$$\mathcal{E}_1 := \left\{ \forall (s,a) \in \mathcal{S} \times \mathcal{A}, \forall i \in [K], \forall h \in [H] \text{ with } k \in [T] : \right.$$
$$\left. |\widehat{R}_{i,h}^k(s,a|\pi_{[i-1]}^k) - R_{i,h}^k(s,a|\pi_{[i-1]}^k)| \leq \frac{\epsilon}{KH} \right\} \tag{9}$$

$$\mathcal{E}_2 := \left\{ \forall (s,a,s') \in \mathcal{S} \times \mathcal{A} \times \mathcal{S}, \forall i \in [K], \forall h \in [H] \text{ with } N_{i,h}^k(s,a) \geq 1 : \right.$$
$$\left. |P_{i,h}(s'|s,a) - \widehat{P}_{i,h}^k(s'|s,a)| \leq \sqrt{\frac{2P_{i,h}(s'|s,a)\iota}{N_{i,h}^k(s,a)}} + \frac{3\iota}{N_{i,h}^k(s,a)} \right\} \tag{10}$$

**Lemma 6.** *The event $\mathcal{E}$ holds with probability $1 - \frac{2\delta}{3}$.*

*Proof.* For any $i \in [K]$, $h \in [H]$, and $l \in [N]$, we consider the following random variable obtained from the $l$-th trajectory $\{(s_{h,l}^j, a_{h,l}^j)\}_{h\in[H],j\in[K]}$ for episode $k$.

$$f(\{(s_{h,l}^j, a_{h,l}^j)\}_{j\in[i-1]} \cup \{(s,a)\}) - f(\{(s_{h,l}^j, a_{h,l}^j)\}_{j\in[i-1]}) = \Delta f(\{(s_{h,l}^j, a_{h,l}^j)\}_{j\in[i-1]}, (s,a))$$

Then it follows that

$$\mathbb{E}[f(\{(s_{h,l}^j, a_{h,l}^j)\}_{j\in[i-1]} \cup \{(s,a)\}) - f(\{(s_{h,l}^j, a_{h,l}^j)\}_{j\in[i-1]})] = \mathbb{E}[\Delta f(\{(s_{h,l}^j, a_{h,l}^j)\}_{j\in[i-1]}, (s,a))]$$
$$= R_{i,h}^k(s,a|\pi_{[i-1]})$$

By Hoeffding's inequality, we have that with probability at least $1 - \frac{\delta}{3K|\mathcal{S}||\mathcal{A}|H}$, for any fixed $(s,a) \in \mathcal{S} \times \mathcal{A}$, and $h \in [H]$, it follows that

$$|R_{i,h}^k(s,a|\pi_{[i-1]}) - \widehat{R}_{i,h}^k(s,a|\pi_{[i-1]})| \leq \frac{\epsilon}{KH}$$

By taking the union bound over all $i \in [K]$, $h \in [H]$, $(s,a) \in \mathcal{S} \times \mathcal{A}$, we have that the event $\mathcal{E}_1$ holds with probability at least $1 - \delta/3$.

Next, since

$$\widehat{P}_{i,h}^k(\cdot|s,a) = \frac{\sum_{e=1}^k \mathbb{I}(s_{h,e}^i = s, a_{h,e}^i = a, s_{h+1,e}^i = s')}{\sum_{e=1}^k \mathbb{I}(s_{h,e}^i = s, a_{h,e}^i = a)}$$
$$= \frac{\sum_{e=1}^k \mathbb{I}(s_{h,e}^i = s, a_{h,e}^i = a) \cdot \mathbb{I}(s_{h+1,e}^i = s')}{\sum_{e=1}^k \mathbb{I}(s_{h,e}^i = s, a_{h,e}^i = a)}$$

Note that conditioned on visiting $(s,a)$, the variable $\mathbb{I}(s_{h+1,e}^i = s')$ is a binary random variable with $\mathbb{E}[\mathbb{I}(s_{h+1,e}^i = s')|s_{h,e}^i = s, a_{h,e}^i = a] = P_{i,h}(s'|s,a)$ and $Var[\mathbb{I}(s_{h+1,e}^i = s')|s_{h,e}^i = s, a_{h,e}^i = a] = P_{i,h}(s'|s,a)(1 - P_{i,h}(s'|s,a)) \leq P_{i,h}(s'|s,a)$.

By applying the standard Bernstein's inequality in Lemma 12, and taking union bound over $(s, a, s')$, $i \in [K]$, $k \in [T]$, $h \in [H]$ we can get that event $\mathcal{E}_2$ holds with probability $1 - \frac{\delta}{3}$.

Since $\mathcal{E}$ is the intersection of the above events, we can prove the lemma by taking a union bound again over all of the two events.

$\square$

### D.2. Key Lemmas

For a given policy $\pi$, we define the value function under the estimated transition dynamics $\widehat{P}^k$ for episode $k$ as $\bar{V}^k$:

$$\bar{V}_h^{\pi,k}(\mathbf{s}) = \mathbb{E}_{\widehat{P}^k, \pi} \left[ \sum_{t=h}^{H} \left( r(\mathbf{s}_t, \mathbf{a}_t) + \sum_{i=1}^{K} b(N_{i,t}^k(s_t^i, a_t^i)) \right) \bigg| \mathbf{s}_h = \mathbf{s} \right] \tag{11}$$

and

$$\bar{Q}_h^{\pi,k}(\mathbf{s}, \mathbf{a}) = \mathbb{E}_{\widehat{P}^k, \pi} \left[ \sum_{t=h}^{H} \left( r(\mathbf{s}_t, \mathbf{a}_t) + \sum_{i=1}^{K} b(N_{i,t}^k(s_t^i, a_t^i)) \right) \bigg| \mathbf{s}_h = \mathbf{s}, \mathbf{a}_h = \mathbf{a} \right],$$

where $b(N) = \sqrt{\frac{2SH^2\iota}{N}} + \frac{3SH\iota}{N}$ is the bonus term. In the case where the joint policy is decomposable with the policy of agent $i$ being denoted as $\pi_i$, we can define the following marginal value function under $\widehat{P}^k$ as

$$\bar{V}_h^{\pi_i,k}(s; \pi_{[i-1]}^k) = \mathbb{E}_{\widehat{P}^k} \left[ \sum_{t=h}^{H} \Delta r_i(\mathbf{s}_t, \mathbf{a}_t) + b(N_{i,t}^k(s_t^i, a_t^i)) \bigg| (\mathbf{s}^{[i-1]}, \mathbf{a}^{[i-1]}) \sim \pi_{[i-1]}^k, s_h^i = s \right]$$

$$\bar{Q}_h^{\pi_i,k}(s, a; \pi_{[i-1]}^k) = \mathbb{E}_{\widehat{P}^k} \left[ \sum_{t=h}^{H} \Delta r_i(\mathbf{s}_t, \mathbf{a}_t) + b(N_{i,t}^k(s_t^i, a_t^i)) \bigg| (\mathbf{s}^{[i-1]}, \mathbf{a}^{[i-1]}) \sim \pi_{[i-1]}^k, s_h^i = s, a_h^i = a \right].$$

By telescoping, we have that

$$\bar{V}_1^{\pi,k}(\bar{\mathbf{s}}_1) = \sum_{i=1}^{K} \bar{V}_1^{\pi_i,k}(\bar{s}_1^i; \pi_{[i-1]}^k). \tag{12}$$

Let $\widehat{V}_{i,h}^k(s)$ denote the estimated value function $\widehat{V}_{i,h}(s)$ for agent $i$ at step $h$ in episode $k$ in Algorithm 2, and let $\pi_i^*$ denote the marginal policy of agent $i$ under the global optimal policy $\pi^*$.

**Lemma 7.** *Conditioned on the event $\mathcal{E}$ in Lemma 6, for all $i \in [K]$, $h \in [H]$, $s \in \mathcal{S}$, $a \in \mathcal{A}$:*

$$\bar{V}_h^{\pi_i^*,k}(s; \pi_{[i-1]}^k) \le \widehat{V}_{i,h}^k(s) \le \bar{V}_h^{\pi_i^k,k}(s; \pi_{[i-1]}^k) + \frac{2\epsilon(H - h + 1)}{KH}$$

*and*

$$\bar{Q}_h^{\pi_i^*,k}(s, a; \pi_{[i-1]}^k) \le \widehat{Q}_{i,h}^k(s, a) \le \bar{Q}_h^{\pi_i^k,k}(s, a; \pi_{[i-1]}^k) + \frac{2\epsilon(H - h + 1)}{KH}.$$

*Proof.* By applying Bellman's equation (Lemma 1) with transition dynamics $\widehat{P}^k$, we have

$$\bar{V}_h^{\pi_i,k}(s; \pi_{[i-1]}^k) = \sum_{a \in \mathcal{A}} \pi_{i,h}(s, a) \bar{Q}_h^{\pi_i,k}(s, a; \pi_{[i-1]}^k)$$

$$\bar{Q}_h^{\pi_i,k}(s, a; \pi_{[i-1]}^k) = R_{i,h}^k(s, a | \pi_{[i-1]}^k) + b(N_{i,h}^k(s, a)) + \sum_{s' \in \mathcal{S}} \widehat{P}_{i,h}^k(s' | s, a) \bar{V}_{h+1}^{\pi_i,k}(s'; \pi_{[i-1]}^k).$$

**Part 1: Proof by induction on $h$ (backward from $H$ to 1).**

*Base case ($h = H$):* By definition,

$$\bar{Q}_H^{\pi_i^*,k}(s,a;\pi_{[i-1]}^k) = \bar{Q}_H^{\pi_i^k,k}(s,a;\pi_{[i-1]}^k) = R_{i,H}^k(s,a|\pi_{[i-1]}^k) + b(N_{i,H}^k(s,a))$$

and

$$\widehat{Q}_{i,H}^k(s,a) = \widehat{R}_{i,H}^k(s,a|\pi_{[i-1]}^k) + b(N_{i,H}^k(s,a)) + \frac{\epsilon}{KH}.$$

Conditioned on the event $\mathcal{E}_1$ in Lemma 6, we have

$$\bar{Q}_H^{\pi_i^*,k}(s,a;\pi_{[i-1]}^k) \le \widehat{Q}_{i,H}^k(s,a) \le \bar{Q}_H^{\pi_i^k,k}(s,a;\pi_{[i-1]}^k) + \frac{2\epsilon}{KH}. \tag{13}$$

For the value function, by Bellman's equation we have $\bar{V}_H^{\pi_i^*,k}(s;\pi_{[i-1]}^k) = \sum_{a\in\mathcal{A}} \pi_{i,H}^*(s,a)\bar{Q}_H^{\pi_i^*,k}(s,a;\pi_{[i-1]}^k)$. Since $\sum_{a\in\mathcal{A}} \pi_{i,H}^*(s,a) = 1$,

$$\begin{aligned}
\bar{V}_H^{\pi_i^*,k}(s;\pi_{[i-1]}^k) &\le \max_{a\in\mathcal{A}} \bar{Q}_H^{\pi_i^*,k}(s,a;\pi_{[i-1]}^k) \\
&\le \max_{a\in\mathcal{A}} \widehat{Q}_{i,H}^k(s,a) && \text{(by (13))} \\
&= \widehat{V}_{i,H}^k(s). && \text{(by definition)}
\end{aligned}$$

By definition of the output policy, $\widehat{V}_{i,H}^k(s) = \widehat{Q}_{i,H}^k(s,\pi_{i,H}^k(s))$ and $\bar{V}_H^{\pi_i^k,k}(s;\pi_{[i-1]}^k) = \bar{Q}_H^{\pi_i^k,k}(s,\pi_{i,H}^k(s);\pi_{[i-1]}^k)$. Thus

$$\begin{aligned}
\widehat{V}_{i,H}^k(s) &= \widehat{Q}_{i,H}^k(s,\pi_{i,H}^k(s)) \\
&\le \bar{Q}_H^{\pi_i^k,k}(s,\pi_{i,H}^k(s);\pi_{[i-1]}^k) + \frac{2\epsilon}{KH} && \text{(by (13))} \\
&= \bar{V}_H^{\pi_i^k,k}(s;\pi_{[i-1]}^k) + \frac{2\epsilon}{KH}.
\end{aligned}$$

This establishes the base case.

*Inductive step:* Assume the result holds for step $h+1$. For step $h$, by Bellman's equation,

$$\begin{aligned}
\bar{Q}_h^{\pi_i^*,k}(s,a;\pi_{[i-1]}^k) &= R_{i,h}^k(s,a|\pi_{[i-1]}^k) + b(N_{i,h}^k(s,a)) + \sum_{s'\in\mathcal{S}} \widehat{P}_{i,h}^k(s'|s,a)\bar{V}_{h+1}^{\pi_i^*,k}(s';\pi_{[i-1]}^k) \\
&\le R_{i,h}^k(s,a|\pi_{[i-1]}^k) + b(N_{i,h}^k(s,a)) + \sum_{s'\in\mathcal{S}} \widehat{P}_{i,h}^k(s'|s,a)\widehat{V}_{i,h+1}^k(s') \\
&\le \widehat{R}_{i,h}^k(s,a|\pi_{[i-1]}^k) + b(N_{i,h}^k(s,a)) + \frac{\epsilon}{KH} + \sum_{s'\in\mathcal{S}} \widehat{P}_{i,h}^k(s'|s,a)\widehat{V}_{i,h+1}^k(s') && \text{(by } \mathcal{E}_1\text{)} \\
&= \widehat{Q}_{i,h}^k(s,a), && \text{(by definition)}
\end{aligned}$$

where the first inequality follows from the assumption that the result holds for $h+1$. For the upper bound, by the inductive hypothesis and Bellman's equation,

$$\begin{aligned}
\widehat{Q}_{i,h}^k(s,a) &= \widehat{R}_{i,h}^k(s,a|\pi_{[i-1]}^k) + b(N_{i,h}^k(s,a)) + \frac{\epsilon}{KH} + \sum_{s'\in\mathcal{S}} \widehat{P}_{i,h}^k(s'|s,a)\widehat{V}_{i,h+1}^k(s') \\
&\le R_{i,h}^k(s,a|\pi_{[i-1]}^k) + b(N_{i,h}^k(s,a)) + \frac{2\epsilon}{KH} \\
&\quad + \sum_{s'\in\mathcal{S}} \widehat{P}_{i,h}^k(s'|s,a)\left\{\bar{V}_{h+1}^{\pi_i^k,k}(s';\pi_{[i-1]}^k) + \frac{2\epsilon(H-h)}{KH}\right\} && \text{(by IH and } \mathcal{E}_1\text{)} \\
&= \bar{Q}_h^{\pi_i^k,k}(s,a;\pi_{[i-1]}^k) + \frac{2\epsilon}{KH} + \frac{2\epsilon(H-h)}{KH} \\
&= \bar{Q}_h^{\pi_i^k,k}(s,a;\pi_{[i-1]}^k) + \frac{2\epsilon(H-h+1)}{KH}.
\end{aligned}$$

For the value functions, by Bellman's equation,

$$\bar{V}_h^{\pi_i^*,k}(s;\pi_{[i-1]}^k) = \sum_{a\in\mathcal{A}} \pi_{i,h}^*(s,a)\bar{Q}_h^{\pi_i^*,k}(s,a;\pi_{[i-1]}^k)$$

$$\leq \max_{a\in\mathcal{A}}\bar{Q}_h^{\pi_i^*,k}(s,a;\pi_{[i-1]}^k)$$

$$\leq \max_{a\in\mathcal{A}}\widehat{Q}_{i,h}^k(s,a) = \widehat{V}_{i,h}^k(s).$$

Since $\widehat{V}_{i,h}^k(s) = \widehat{Q}_{i,h}^k(s,\pi_{i,h}^k(s))$,

$$\widehat{V}_{i,h}^k(s) = \widehat{Q}_{i,h}^k(s,\pi_{i,h}^k(s))$$

$$\leq \bar{Q}_h^{\pi_i^k,k}(s,\pi_{i,h}^k(s);\pi_{[i-1]}^k) + \frac{2\epsilon(H-h+1)}{KH}$$

$$= \bar{V}_h^{\pi_i^k,k}(s;\pi_{[i-1]}^k) + \frac{2\epsilon(H-h+1)}{KH}.$$

This completes the proof. $\qquad\square$

Next, we prove the following lemma, which guarantees that the output policy achieves a $1/2$-approximation ratio with respect to the empirical transition dynamics approximately.

**Lemma 8.** *Throughout the execution of Algorithm 2, we have that for any $k\in[T]$, $\bar{V}_1^{\pi^k,k}(\bar{\mathbf{s}}_1) \geq \frac{\bar{V}_1^{\pi^*,k}(\mathbf{s}_1)}{2} - \epsilon$.*

*Proof.* Suppose we have two groups of $K$ agents, the first group of $K$ agents follows the policy $\pi^k$ executed in by Algorithm 2 at episode $k$ and transition dynamics $\hat{P}^k$. We denote the trajectory by $\{(s_h^i,a_h^i)\}_{h\in[H],i\in[K]}$. The second group of $K$ agents follows the optimal policy $\pi^*$ and transition dynamics $\hat{P}^k$, and we denote the trajectory of the second group of agents by $\{(\tilde{s}_h^i,\tilde{a}_h^i)\}_{h\in[H],i\in[K]}$. In addition, we define the function $\widehat{f}^k: 2^{\mathcal{S}\times\mathcal{A}}\to\mathbb{R}$ as $\widehat{f}^k(S) = f(S) + \sum_{(s,a)\in S} b(N_{i,h}^k(s,a))$. Since $\sum_{(s,a)\in S} b(N_{i,h}^k(s,a))$ is a linear function, and b(N) is positive for each $N\geq 0$, $\widehat{f}^k(S)$ is also monotone and submodular. For notation simplicity, we define

$$G_i^k = \mathbb{E}_{\hat{P}^k}[\sum_{h=1}^H \Delta\widehat{f}^k(\{(s_h^j,a_h^j)\}_{j\in[K]}\cup\{(\tilde{s}_h^m,\tilde{a}_h^m)\}_{m\in[i-1]},(\tilde{s}_h^i,\tilde{a}_h^i))|(\mathbf{s},\mathbf{a})\sim\pi^k,(\tilde{\mathbf{s}},\tilde{\mathbf{a}})\sim\pi^*], \tag{14}$$

where $(\mathbf{s},\mathbf{a})\sim\pi^k,(\tilde{\mathbf{s}},\tilde{\mathbf{a}})\sim\pi^*$ means that $\{(s_h^i,a_h^i)\}_{h\in[H],i\in[K]}$ is sampled according to policy $\pi^k$, and $\{(\tilde{s}_h^i,\tilde{a}_h^i)\}_{h\in[H],i\in[K]}$ is sampled based on optimal policy $\pi^*$. Then by definition, we have that

$$G_i^k = \mathbb{E}_{\hat{P}^k}[\sum_{h=1}^H \widehat{f}^k\{(s_h^j,a_h^j)\}_{j\in[K]}\cup\{(\tilde{s}_h^m,\tilde{a}_h^m)\}_{m\in[i]} - \widehat{f}^k\{(s_h^j,a_h^j)\}_{j\in[K]}\cup\{(\tilde{s}_h^m,\tilde{a}_h^m)\}_{m\in[i-1]})|(\mathbf{s},\mathbf{a})\sim\pi^k,(\tilde{\mathbf{s}},\tilde{\mathbf{a}})\sim\pi^*].$$

It then follows that

$$\sum_{i=1}^K G_i^k = \mathbb{E}_{\hat{P}^k}[\sum_{h=1}^H \widehat{f}^k\{(s_h^j,a_h^j)\}_{j\in[K]}\cup\{(\tilde{s}_h^m,\tilde{a}_h^m)\}_{m\in[K]} - \widehat{f}^k\{(s_h^j,a_h^j)\}_{j\in[K]})|(\mathbf{s},\mathbf{a})\sim\pi^k,(\tilde{\mathbf{s}},\tilde{\mathbf{a}})\sim\pi^*]$$

$$\geq \mathbb{E}_{\hat{P}^k}[\sum_{h=1}^H \widehat{f}^k\{(\tilde{s}_h^m,\tilde{a}_h^m)\}_{m\in[K]} - \widehat{f}^k\{(s_h^j,a_h^j)\}_{j\in[K]})|(\mathbf{s},\mathbf{a})\sim\pi^k,(\tilde{\mathbf{s}},\tilde{\mathbf{a}})\sim\pi^*]$$

$$= \mathbb{E}_{\hat{P}^k}[\sum_{h=1}^H \widehat{f}^k\{(\tilde{s}_h^m,\tilde{a}_h^m)\}_{m\in[K]})|(\tilde{\mathbf{s}},\tilde{\mathbf{a}})\sim\pi^*] - \mathbb{E}_{\hat{P}^k}[\sum_{h=1}^H \widehat{f}^k\{(s_h^j,a_h^j)\}_{j\in[K]})|(\mathbf{s},\mathbf{a})\sim\pi^k]$$

$$= \bar{V}_1^{\pi^*,k}(\bar{\mathbf{s}}_1) - \bar{V}_1^{\pi^k,k}(\bar{\mathbf{s}}_1), \tag{15}$$

where the inequality follows from the fact that $f$ is monotone, and the last inequality follows from definition of $\bar{V}_1^{\pi,k}(\bar{\mathbf{s}}_1)$ as presented in (11). Next, we bound $G_i^k$ for each $i$. Since $\{(s_h^l, a_h^l)\}_{l \in [i-1]} \subseteq \{(s_h^j, a_h^j)\}_{j \in [K]} \cup \{(\tilde{s}_h^m, \tilde{a}_h^m)\}_{m \in [i]}$, by submodularity, we have that

$$G_i^k \leq \mathbb{E}[\sum_{h=1}^{H} \Delta \widehat{f}^k(\{(s_h^l, a_h^l)\}_{l \in [i-1]}, (\tilde{s}_h^i, \tilde{a}_h^i))|(\mathbf{s}, \mathbf{a}) \sim \pi^k, (\tilde{\mathbf{s}}, \tilde{\mathbf{a}}) \sim \pi^*]. \tag{16}$$

Let us denote the policy $\pi_{i,h}^*$ to be the marginal policy of the $i$-th agent at time step $h$ while the global policy is $\pi^*$. For notational simplicity, here we also use $\pi_i^* = \{\pi_{i,h}^*\}_{h \in [H]}$ to denote the marginal policy of agent $i$ throughout the entire episode. Therefore,

$$\mathbb{E}[\sum_{h=1}^{H} \Delta \widehat{f}^k(\{(s_h^l, a_h^l)\}_{l \in [i-1]}, (\tilde{s}_h^i, \tilde{a}_h^i))|(\mathbf{s}, \mathbf{a}) \sim \pi^k, (\tilde{\mathbf{s}}, \tilde{\mathbf{a}}) \sim \pi^*]$$
$$= \mathbb{E}[\sum_{h=1}^{H} \Delta \widehat{f}^k(\{(s_h^l, a_h^l)\}_{l \in [i-1]}, (\tilde{s}_h^i, \tilde{a}_h^i))|(\mathbf{s}, \mathbf{a}) \sim \pi^k, (\tilde{s}^i, \tilde{a}^i) \sim \pi_i^*]$$
$$= \bar{V}_1^{\pi_i^*, k}(\bar{s}_1; \pi_{[i-1]}).$$

Combine the above inequality with (16), it then follows that

$$G_i^k \leq \bar{V}_1^{\pi_i^*, k}(\bar{s}_1; \pi_{[i-1]}). \tag{17}$$

Next, from the result in Lemma 7, we would have that

$$G_i^k \leq \bar{V}_1^{\pi_i^k, k}(\bar{s}_1^i; \pi_{[i-1]}) + 2\frac{\epsilon}{K}$$

Since $\sum_{i=1}^{K} G_i^k \geq \bar{V}_1^{\pi^*, k}(\bar{\mathbf{s}}_1) - \bar{V}_1^{\pi^k, k}(\bar{\mathbf{s}}_1)$ from (15), we have that

$$\bar{V}_1^{\pi^*, k}(\bar{\mathbf{s}}_1) - \bar{V}_1^{\pi^k, k}(\bar{\mathbf{s}}_1) \leq \sum_{i=1}^{K} \{\bar{V}_1^{\pi_i^k, k}(\bar{s}_1^i; \pi_{[i-1]}) + \frac{2\epsilon}{K}\}$$
$$= \bar{V}_1^{\pi^k, k}(\bar{\mathbf{s}}_1) + 2\epsilon.$$

where the last equality follows from the value decomposition in (12). By rearranging the inequality above, we can prove the result in the theorem. $\qquad\square$

Next, we prove the following decomposition lemma.

**Lemma 9.** *For any joint state-action pair* $(\mathbf{s}, \mathbf{a}) \in \mathcal{S}^K \times \mathcal{A}^K$ *and any function* $G : \mathcal{S}^K \to \mathbb{R}$ *satisfying* $0 \leq G(\mathbf{s}) \leq H$ *for all* $\mathbf{s} \in \mathcal{S}^K$, *we have*

$$\sum_{\mathbf{s}' \in \mathcal{S}^K} [\widehat{\mathbf{P}}_h^k(\mathbf{s}'|\mathbf{s}, \mathbf{a}) - \mathbf{P}_h(\mathbf{s}'|\mathbf{s}, \mathbf{a})]G(\mathbf{s}') \leq \frac{1}{H} \sum_{\mathbf{s}' \in \mathcal{S}^K} \mathbf{P}_h(\mathbf{s}'|\mathbf{s}, \mathbf{a})G(\mathbf{s}') + \sum_{i=1}^{K} \frac{10SH^2K\iota}{N_{i,h}^k(s^i, a^i)}.$$

*Proof.* The proof exploits the product structure of multi-agent transition dynamics through a telescoping argument over agents, combined with carefully calibrated applications of concentration inequalities.

**Step 1: Telescoping decomposition.** By the independence of agent transitions, the joint transition probability factors as:

$$\widehat{\mathbf{P}}_h^k(\mathbf{s}'|\mathbf{s}, \mathbf{a}) = \prod_{i=1}^{K} \widehat{P}_{i,h}^k(s'^i|s^i, a^i), \quad \mathbf{P}_h(\mathbf{s}'|\mathbf{s}, \mathbf{a}) = \prod_{i=1}^{K} P_{i,h}(s'^i|s^i, a^i).$$

Our target quantity becomes:

$$\sum_{\mathbf{s}'\in\mathcal{S}^K}\widehat{\mathbf{P}}_h^k(\mathbf{s}'|\mathbf{s},\mathbf{a})G(\mathbf{s}') - \sum_{\mathbf{s}'\in\mathcal{S}^K}\mathbf{P}_h(\mathbf{s}'|\mathbf{s},\mathbf{a})G(\mathbf{s}')$$

$$= \sum_{\mathbf{s}'\in\mathcal{S}^K}\left[\prod_{i=1}^K\widehat{P}_{i,h}^k(s'^i|s^i,a^i) - \prod_{i=1}^K P_{i,h}(s'^i|s^i,a^i)\right]G(\mathbf{s}'). \tag{18}$$

To decompose this difference of products, we introduce a sequence of intermediate terms that gradually transitions from using all true probabilities to using all empirical probabilities. For $i \in \{0, 1, \ldots, K\}$, define:

$$Y_i := \sum_{\mathbf{s}'\in\mathcal{S}^K}\left[\prod_{l=1}^i\widehat{P}_{l,h}^k(s'^l|s^l,a^l)\right]\left[\prod_{m=i+1}^K P_{m,h}(s'^m|s^m,a^m)\right]G(\mathbf{s}'), \tag{19}$$

where we adopt the convention that empty products equal 1. This construction allows us to bridge the gap between empirical and true expectations by changing one agent's transition at a time. By telescoping, we can write the total difference as a sum of incremental changes:

$$\sum_{\mathbf{s}'\in\mathcal{S}^K}\widehat{\mathbf{P}}_h^k(\mathbf{s}'|\mathbf{s},\mathbf{a})G(\mathbf{s}') - \sum_{\mathbf{s}'\in\mathcal{S}^K}\mathbf{P}_h(\mathbf{s}'|\mathbf{s},\mathbf{a})G(\mathbf{s}') = Y_K - Y_0 = \sum_{i=1}^K(Y_i - Y_{i-1}). \tag{20}$$

Each difference $Y_i - Y_{i-1}$ captures the effect of replacing agent $i$'s true transition probability with its empirical estimate, while keeping all other agents' transitions fixed. This isolates individual agent errors and enables us to apply concentration inequalities separately for each agent.

**Step 2: Bounding individual agent contributions.** By definition (19), the terms $Y_i$ and $Y_{i-1}$ differ only in the transition probability for agent $i$:

$$Y_i - Y_{i-1} = \sum_{\mathbf{s}'\in\mathcal{S}^K}\prod_{l=1}^{i-1}\widehat{P}_{l,h}^k(s'^l|s^l,a^l)\prod_{m=i+1}^K P_{m,h}(s'^m|s^m,a^m)G(\mathbf{s}')(\widehat{P}_{i,h}^k(s'^i|s^i,a^i) - P_{i,h}(s'^i|s^i,a^i))$$

$$\leq \sum_{\mathbf{s}'\in\mathcal{S}^K}\prod_{l=1}^{i-1}\widehat{P}_{l,h}^k(s'^l|s^l,a^l)\prod_{m=i+1}^K P_{m,h}(s'^m|s^m,a^m)G(\mathbf{s}')|\widehat{P}_{i,h}^k(s'^i|s^i,a^i) - P_{i,h}(s'^i|s^i,a^i)|$$

$$= \sum_{s'\in\mathcal{S}}|\widehat{P}_{i,h}^k(s'|s^i,a^i) - P_{i,h}(s'|s^i,a^i)|\sum_{\mathbf{s}':s'^i=s'}\prod_{l=1}^{i-1}\widehat{P}_{l,h}^k(s'^l|s^l,a^l)\prod_{m=i+1}^K P_{m,h}(s'^m|s^m,a^m)G(\mathbf{s}').$$

In addition, let us denote $Z_i(s') = \sum_{\mathbf{s}':s'^i=s'}\prod_{l=1}^{i-1}\widehat{P}_{l,h}^k(s'^l|s^l,a^l)\prod_{m=i+1}^K P_{m,h}(s'^m|s^m,a^m)G(\mathbf{s}')$. Then

$$Y_i - Y_{i-1} \leq \sum_{s'\in\mathcal{S}}|\widehat{P}_{i,h}^k(s'|s^i,a^i) - P_{i,h}(s'|s^i,a^i)|Z_i(s').$$

By event $\mathcal{E}_2$ in Lemma 6, we have that

$$\sum_{s'\in\mathcal{S}}|\widehat{P}_{i,h}^k(s'|s^i,a^i) - P_{i,h}(s'|s^i,a^i)|Z_i(s') \leq \sum_{s'\in\mathcal{S}}\left(\sqrt{\frac{2P_{i,h}(s'|s^i,a^i)\iota}{N_{i,h}^k(s^i,a^i)}} + \frac{3\iota}{N_{i,h}^k(s^i,a^i)}\right)Z_i(s')$$

$$\leq \sum_{s'\in\mathcal{S}}\left(\frac{P_{i,h}(s'|s^i,a^i)}{2HK} + \frac{HK\iota}{N_{i,h}^k(s^i,a^i)} + \frac{3\iota}{N_{i,h}^k(s^i,a^i)}\right)Z_i(s')$$

$$\leq \frac{\sum_{s'\in\mathcal{S}}P_{i,h}(s'|s^i,a^i)Z_i(s')}{2HK} + \frac{4HK\iota}{N_{i,h}^k(s^i,a^i)}\sum_{s'\in\mathcal{S}}Z_i(s'), \tag{21}$$

where the second inequality follows from $\sqrt{ab} \leq \frac{a+b}{2}$ with $a = \frac{P_{i,h}(s'|s^i,a^i)}{2HK}$ and $b = \frac{2HK\iota}{N_{i,h}^k(s^i,a^i)}$.

Since $G(\mathbf{s}') \leq H$ by assumption, we can bound the $Z_i(s')$ by factoring out $H$:

$$Z_i(s') \leq H \sum_{\mathbf{s}':s'^i=s'} \prod_{l=1}^{i-1} \widehat{P}_{l,h}^k(s'^l|s^l,a^l) \prod_{m=i+1}^{K} P_{m,h}(s'^m|s^m,a^m) = H,$$

By definition, we have that

$$\sum_{s'\in\mathcal{S}} P_{i,h}(s'|s^i,a^i)Z_i(s') = \sum_{\mathbf{s}'\in\mathcal{S}^K} \left[\prod_{l=1}^{i-1} \widehat{P}_{l,h}^k(s'^l|s^l,a^l)\right] \left[\sum_{s'^i\in\mathcal{S}} P_{i,h}(s'^i|s^i,a^i)\right] \left[\prod_{m=i+1}^{K} P_{m,h}(s'^m|s^m,a^m)\right] G(\mathbf{s}')$$

$$= \sum_{\mathbf{s}'\in\mathcal{S}^K} \left[\prod_{l=1}^{i-1} \widehat{P}_{l,h}^k(s'^l|s^l,a^l)\right] \left[\prod_{m=i}^{K} P_{m,h}(s'^m|s^m,a^m)\right] G(\mathbf{s}') = Y_{i-1}, \tag{22}$$

Combining with (21), we can get

$$Y_i - Y_{i-1} \leq \frac{Y_{i-1}}{2HK} + \frac{4HK\iota}{N_{i,h}^k(s^i,a^i)} \sum_{s'\in\mathcal{S}} Z_i(s')$$

$$\leq \frac{Y_{i-1}}{2HK} + \frac{4H^2SK\iota}{N_{i,h}^k(s^i,a^i)}. \tag{23}$$

From (20) and (23), we get

$$\sum_{\mathbf{s}'} \widehat{\mathbf{P}}_h^k(\mathbf{s}'|\mathbf{s},\mathbf{a})G(\mathbf{s}') - \sum_{\mathbf{s}'} \mathbf{P}_h(\mathbf{s}'|\mathbf{s},\mathbf{a})G(\mathbf{s}') \leq \frac{1}{2HK} \sum_{i=1}^{K} Y_{i-1} + \sum_{i=1}^{K} \frac{4SH^2K\iota}{N_{i,h}^k(s^i,a^i)}. \tag{24}$$

To bound $\sum_{i=1}^{K} Y_{i-1}$, we require an upper bound on each $Y_i$.

**Step 3: Refined recursive bound on $Y_i$.**  Next, we bound $Y_i$ recursively. Using a similar argument with $a = \frac{P_{i,h}(s'|s^i,a^i)}{K}$ and $b = \frac{K\iota}{N_{i,h}^k(s^i,a^i)}$, we obtain

$$\sum_{s'\in\mathcal{S}} |\widehat{P}_{i,h}^k(s'|s^i,a^i) - P_{i,h}(s'|s^i,a^i)| \leq \frac{\sum_{s'\in\mathcal{S}} P_{i,h}(s'|s^i,a^i)}{K} + \frac{4K\iota}{N_{i,h}^k(s^i,a^i)}.$$

Similarly as in Step 2, we can show

$$Y_i - Y_{i-1} \leq \sum_{s'\in\mathcal{S}} |\widehat{P}_{i,h}^k(s'|s^i,a^i) - P_{i,h}(s'|s^i,a^i)|Z_i(s')$$

$$\leq \frac{\sum_{s'\in\mathcal{S}} P_{i,h}(s'|s^i,a^i)Z_i(s')}{K} + \frac{4HK\iota}{N_{i,h}^k(s^i,a^i)} \sum_{s'\in\mathcal{S}} Z_i(s')$$

$$\leq \frac{1}{K}Y_{i-1} + \frac{4HSK\iota}{N_{i,h}^k(s^i,a^i)},$$

which gives

$$Y_i \leq (1 + \frac{1}{K})Y_{i-1} + \frac{4HSK\iota}{N_{i,h}^k(s^i,a^i)}.$$

By recursion, we obtain

$$Y_i \leq (1 + \frac{1}{K})^i Y_0 + \sum_{m=1}^{i} \frac{4SHK\iota}{N_{m,h}^k(s^m,a^m)}(1 + \frac{1}{K})^{i-m}.$$

Therefore,

$$
\begin{aligned}
\sum_{i=1}^{K} Y_{i-1} &\leq \sum_{i=1}^{K}(1+\frac{1}{K})^{i-1}Y_0 + \sum_{i=2}^{K}\sum_{m=1}^{i-1}\frac{4SHK\iota}{N_{m,h}^k(s^m,a^m)}(1+\frac{1}{K})^{i-1-m} \\
&\leq \frac{(1+\frac{1}{K})^K - 1}{\frac{1}{K}}Y_0 + \sum_{m=1}^{K-1}\sum_{i=m+1}^{K}\frac{4SHK\iota}{N_{m,h}^k(s^m,a^m)}(1+\frac{1}{K})^{i-1-m} \\
&\leq K[(1+\frac{1}{K})^K - 1]Y_0 + \sum_{m=1}^{K-1}\frac{4SHK\iota}{N_{m,h}^k(s^m,a^m)}\sum_{j=0}^{K-m-1}(1+\frac{1}{K})^j \\
&\leq K(e-1)Y_0 + \sum_{m=1}^{K-1}\frac{4SHK^2\iota}{N_{m,h}^k(s^m,a^m)}e,
\end{aligned}
\tag{25}
$$

where we used $(1+\frac{1}{K})^K \leq e$ and $\sum_{j=0}^{K}(1+\frac{1}{K})^j \leq K\left((1+\frac{1}{K})^K - 1\right)$.

**Step 4: Final Bound.** Substituting (25) into (24):

$$
\begin{aligned}
&\sum_{\mathbf{s}'}\widehat{\mathbf{P}}_h^k(\mathbf{s}'|\mathbf{s},\mathbf{a})G(\mathbf{s}') - \sum_{\mathbf{s}'}\mathbf{P}_h(\mathbf{s}'|\mathbf{s},\mathbf{a})G(\mathbf{s}') \\
&\leq \frac{e-1}{2H}Y_0 + 2SKe\iota\sum_{m=1}^{K-1}\frac{1}{N_{m,h}^k(s^m,a^m)} + \sum_{i=1}^{K}\frac{4SH^2K\iota}{N_{i,h}^k(s^i,a^i)}.
\end{aligned}
$$

Since $e < 3$, we have $\frac{e-1}{2H} < \frac{1}{H}$ and $2SKe\iota < 6SK\iota$. Extending the first sum to include $i = K$ and combining:

$$
\begin{aligned}
&\sum_{\mathbf{s}'}\widehat{\mathbf{P}}_h^k(\mathbf{s}'|\mathbf{s},\mathbf{a})G(\mathbf{s}') - \sum_{\mathbf{s}'}\mathbf{P}_h(\mathbf{s}'|\mathbf{s},\mathbf{a})G(\mathbf{s}') \\
&\leq \frac{1}{H}Y_0 + \sum_{i=1}^{K}\frac{6SK\iota + 4SH^2K\iota}{N_{i,h}^k(s^i,a^i)} \\
&\leq \frac{1}{H}Y_0 + \sum_{i=1}^{K}\frac{10SH^2K\iota}{N_{i,h}^k(s^i,a^i)}.
\end{aligned}
$$

Thus we complete the proof. □

### D.3. Optimistic Estimate

Let us recall the definition of $\bar{V}_h^{\pi,k}(\mathbf{s})$, which is

$$
\bar{V}_h^{\pi,k}(\mathbf{s}) = \mathbb{E}_{\widehat{P}^k,\pi}\left[\sum_{t=h}^{H}\left(r(\mathbf{s}_t,\mathbf{a}_t) + \sum_{i=1}^{K}b(N_{i,t}^k(s_t^i,a_t^i))\right)\bigg|\mathbf{s}_h = \mathbf{s}\right].
$$

In this section, we prove that $\bar{V}_h^{\pi,k}(\mathbf{s})$ is indeed an optimistic estimate of the true value function $V_h^\pi$ for any policy $\pi$. First of all, we prove the following lemma.

**Lemma 10.** *Conditioned on event $\mathcal{E}$ (Lemma 6), for any $\mathbf{s} \in \mathcal{S}^K$, $\mathbf{a} \in \mathcal{A}^K$, $h \in [H]$, $k \in [T]$, and any function $V : \mathcal{S}^K \to \mathbb{R}$ with $V(\mathbf{s}) \in [0, H]$,*

$$
\left|\sum_{\mathbf{s}'}\widehat{\mathbf{P}}_h^k(\mathbf{s}'|\mathbf{s},\mathbf{a})V(\mathbf{s}') - \sum_{\mathbf{s}'}\mathbf{P}_h(\mathbf{s}'|\mathbf{s},\mathbf{a})V(\mathbf{s}')\right| \leq \sum_{i=1}^{K}b(N_{i,h}^k(s^i,a^i)),
$$

*where $b(N) = H\sqrt{\frac{2S\iota}{N}} + \frac{3SH\iota}{N}$ and $\iota = \log(6S^2ATHK/\delta)$.*

*Proof.* The proof of the lemma is similar to that of Lemma 9. By independence of agent transitions, $\mathbf{P}_h(\mathbf{s}'|\mathbf{s}, \mathbf{a}) = \prod_{i=1}^{K} P_{i,h}(s'^i|s^i, a^i)$ and similarly for $\widehat{\mathbf{P}}_h^k$. Define

$$X_i := \sum_{\mathbf{s}'} \prod_{l=1}^{i} \widehat{P}_{l,h}^k(s'^l|s^l, a^l) \prod_{m=i+1}^{K} P_{m,h}(s'^m|s^m, a^m) V(\mathbf{s}'), \quad i \in \{0, \ldots, K\},$$

where empty products equal 1. Note $X_0$ uses all true transitions and $X_K$ uses all empirical transitions. By telescoping,

$$\left| \sum_{\mathbf{s}'} \widehat{\mathbf{P}}_h^k V(\mathbf{s}') - \sum_{\mathbf{s}'} \mathbf{P}_h V(\mathbf{s}') \right| = |X_K - X_0| \leq \sum_{i=1}^{K} |X_i - X_{i-1}|. \tag{26}$$

For each $i \in [K]$, $X_i$ and $X_{i-1}$ differ only in agent $i$'s transition. Rearranging,

$$X_i - X_{i-1} = \sum_{s'^i} \left( \widehat{P}_{i,h}^k(s'^i|s^i, a^i) - P_{i,h}(s'^i|s^i, a^i) \right) \cdot W_i(s'^i),$$

where $W_i(s'^i) := \sum_{\mathbf{s}':\, s'^i \text{ fixed}} \prod_{l \neq i} \mathbb{P}_{l,h}(s'^l|s^l, a^l) V(\mathbf{s}') \in [0, H]$ and $\mathbb{P}_{l,h}$ denotes $\widehat{P}$ for $l < i$ and $P$ for $l > i$. Since $|W_i(s'^i)| \leq H$ (probabilities sum to 1),

$$|X_i - X_{i-1}| \leq H \sum_{s' \in \mathcal{S}} |\widehat{P}_{i,h}^k(s'|s^i, a^i) - P_{i,h}(s'|s^i, a^i)|. \tag{27}$$

By event $\mathcal{E}_2$ (Lemma 6),

$$|\widehat{P}_{i,h}^k(s'|s^i, a^i) - P_{i,h}(s'|s^i, a^i)| \leq \sqrt{\frac{2 P_{i,h}(s'|s^i, a^i) \iota}{N_{i,h}^k(s^i, a^i)}} + \frac{3\iota}{N_{i,h}^k(s^i, a^i)}.$$

Summing over $s' \in \mathcal{S}$ and applying Cauchy-Schwarz inequality,

$$\sum_{s'} |\widehat{P}_{i,h}^k(s'|s^i, a^i) - P_{i,h}(s'|s^i, a^i)| \leq \sqrt{\frac{2\iota}{N_{i,h}^k(s^i, a^i)}} \underbrace{\sum_{s'} \sqrt{P_{i,h}(s'|s^i, a^i)}}_{\leq \sqrt{S}} + \frac{3S\iota}{N_{i,h}^k(s^i, a^i)}$$

$$\leq \sqrt{\frac{2S\iota}{N_{i,h}^k(s^i, a^i)}} + \frac{3S\iota}{N_{i,h}^k(s^i, a^i)}.$$

Combining with (27) gives $|X_i - X_{i-1}| \leq b(N_{i,h}^k(s^i, a^i))$. The result follows from (26). □

Next, we prove the following lemma.

**Lemma 11.** *Conditioned on the event $\mathcal{E}$ in Lemma 6, we have that throughout the Algorithm 2, it holds that for any $\mathbf{s} \in \mathcal{S}$, $\mathbf{a} \in \mathcal{A}^K$, and any policy $\pi$, it holds that*

$$\bar{V}_h^{\pi,k}(\mathbf{s}) \geq V_h^{\pi}(\mathbf{s})$$

*and*

$$\bar{Q}_h^{\pi,k}(\mathbf{s}, \mathbf{a}) \geq Q_h^{\pi}(\mathbf{s}, \mathbf{a})$$

*Proof.* First of all, consider the modified Markov Decision Process $\widehat{M}$ under the transition dynamics $\widehat{\mathbf{P}}^k$ with reward being $r(\mathbf{s}, \mathbf{a}) + \sum_{i=1}^{K} b(N_{i,h}^k(s^i, a^i))$, where $b(\cdot) \geq 0$ is the bonus function. By definition, $\bar{V}^{\pi,k}$ and $\bar{Q}^{\pi,k}$ are the value functions

for policy $\pi$ in this modified MDP. By apply the Bellman's equation for $\widehat{M}$, we can get for any $\mathbf{s} = (s^1, s^2, ..., s^K) \in \mathcal{S}^K$ and $\mathbf{a} = (a^1, a^2, ..., a^K) \in \mathcal{A}^K$,

$$\bar{V}_h^{\pi,k}(\mathbf{s}) = \sum_{\mathbf{a} \in \mathcal{A}^K} \pi(\mathbf{s}, \mathbf{a}) \bar{Q}_h^{\pi,k}(\mathbf{s}, \mathbf{a})$$

$$\bar{Q}_h^{\pi,k}(\mathbf{s}, \mathbf{a}) = r(\mathbf{s}, \mathbf{a}) + \sum_{i=1}^{K} b(N_{i,h}^k(s^i, a^i)) + \sum_{\mathbf{s}' \in \mathcal{S}^K} \widehat{\mathbf{P}}_h^k(\mathbf{s}'|\mathbf{s}, \mathbf{a}) \bar{V}_{h+1}^{\pi,k}(\mathbf{s}') \qquad (28)$$

where $\widehat{\mathbf{P}}_h^k(\mathbf{s}'|\mathbf{s}, \mathbf{a}) = \prod_{i=1}^{K} \widehat{P}_{i,h}^k(s'^i|s^i, a^i)$ is the joint transition probability from $(\mathbf{s}, \mathbf{a})$ to $\mathbf{s}'$.

Similarly, for the true value functions:

$$V_h^\pi(\mathbf{s}) = \sum_{\mathbf{a} \in \mathcal{A}^K} \pi(\mathbf{s}, \mathbf{a}) Q_h^\pi(\mathbf{s}, \mathbf{a})$$

$$Q_h^\pi(\mathbf{s}, \mathbf{a}) = r(\mathbf{s}, \mathbf{a}) + \sum_{\mathbf{s}' \in \mathcal{S}^K} \mathbf{P}_h(\mathbf{s}'|\mathbf{s}, \mathbf{a}) V_{h+1}^\pi(\mathbf{s}') \qquad (29)$$

where $\mathbf{P}_h(\mathbf{s}'|\mathbf{s}, \mathbf{a}) = \prod_{i=1}^{K} P_{i,h}(s'^i|s^i, a^i)$.

Next, we prove the result by induction. First of all, when $h = H$, we have that

$$\bar{Q}_H^{\pi,k}(\mathbf{s}, \mathbf{a}) = r(\mathbf{s}, \mathbf{a}) + \sum_{i=1}^{K} b(N_{i,H}^k(s^i, a^i))$$

$$\geq r(\mathbf{s}, \mathbf{a}) = Q_H^\pi(\mathbf{s}, \mathbf{a}).$$

Since $\bar{V}_h^{\pi,k}(\mathbf{s}) = \sum_{\mathbf{a} \in \mathcal{A}^K} \pi(\mathbf{s}, \mathbf{a}) \bar{Q}_h^{\pi,k}(\mathbf{s}, \mathbf{a})$ and $V_h^\pi(\mathbf{s}) = \sum_{\mathbf{a} \in \mathcal{A}^K} \pi(\mathbf{s}, \mathbf{a}) Q_h^\pi(\mathbf{s}, \mathbf{a})$, we have that the results hold for $h = H$.

Next, suppose results hold for $h + 1$. From the Bellman's equation in (28), we have that

$$\bar{Q}_h^{\pi,k}(\mathbf{s}, \mathbf{a}) = r(\mathbf{s}, \mathbf{a}) + \sum_{\mathbf{s}' \in \mathcal{S}^K} \widehat{\mathbf{P}}_h^k(\mathbf{s}'|\mathbf{s}, \mathbf{a}) \bar{V}_{h+1}^{\pi,k}(\mathbf{s}') + \sum_{i=1}^{K} b(N_{i,h}^k(s^i, a^i))$$

$$\geq r(\mathbf{s}, \mathbf{a}) + \sum_{\mathbf{s}' \in \mathcal{S}^K} \widehat{\mathbf{P}}_h^k(\mathbf{s}'|\mathbf{s}, \mathbf{a}) V_{h+1}^\pi(\mathbf{s}') + \sum_{i=1}^{K} b(N_{i,h}^k(s^i, a^i)).$$

From Lemma 10, we can get that

$$\bar{Q}_h^{\pi,k}(\mathbf{s}, \mathbf{a}) \geq r(\mathbf{s}, \mathbf{a}) + \sum_{\mathbf{s}' \in \mathcal{S}^K} \mathbf{P}(\mathbf{s}'|\mathbf{s}, \mathbf{a}) V_{h+1}^\pi(\mathbf{s}')$$

$$\geq Q_h^\pi(\mathbf{s}, \mathbf{a})$$

Since $\bar{V}_h^{\pi,k}(\mathbf{s}) = \sum_{\mathbf{a} \in \mathcal{A}^K} \pi(\mathbf{s}, \mathbf{a}) \bar{Q}_h^{\pi,k}(\mathbf{s}, \mathbf{a}) \geq \sum_{\mathbf{a} \in \mathcal{A}^K} \pi(\mathbf{s}, \mathbf{a}) Q_h^\pi(\mathbf{s}, \mathbf{a}) = V_h^\pi(\mathbf{s})$, the result holds for the case of $h$. Thus, we can conclude the proof. $\qquad\square$

### D.4. Proof of Main Theorem

We now establish our main regret bound. The proof proceeds by relating the regret to the gap between optimistic value estimates and true values, decomposing this gap via a Bellman-style recursion, and bounding the resulting terms using concentration inequalities.

#### D.4.1. RELATING REGRET TO VALUE ESTIMATES

By definition,

$$\mathcal{R}_{T, \frac{1}{2}} = \sum_{k=1}^{T} \left[ \frac{1}{2} V_1^{\pi^*}(\bar{\mathbf{s}}_1) - V_1^{\pi^k}(\bar{\mathbf{s}}_1) \right].$$

From Lemma 11, we have $\bar{V}_1^{\pi^*,k}(\bar{\mathbf{s}}_1) \geq V_1^{\pi^*}(\bar{\mathbf{s}}_1)$ for all $k$, which gives

$$\mathcal{R}_{T,\frac{1}{2}} \leq \sum_{k=1}^{T} \left[ \frac{1}{2} \bar{V}_1^{\pi^*,k}(\bar{\mathbf{s}}_1) - V_1^{\pi^k}(\bar{\mathbf{s}}_1) \right]. \tag{30}$$

From Lemma 8, we have $\bar{V}_1^{\pi^k,k}(\bar{\mathbf{s}}_1) \geq \frac{\bar{V}_1^{\pi^*,k}(\bar{\mathbf{s}}_1)}{2} - \epsilon$, which rearranges to $\frac{1}{2}\bar{V}_1^{\pi^*,k}(\bar{\mathbf{s}}_1) \leq \bar{V}_1^{\pi^k,k}(\bar{\mathbf{s}}_1) + \epsilon$. Substituting into (30):

$$\mathcal{R}_{T,\frac{1}{2}} \leq \sum_{k=1}^{T} \left[ \bar{V}_1^{\pi^k,k}(\bar{\mathbf{s}}_1) - V_1^{\pi^k}(\bar{\mathbf{s}}_1) + \epsilon \right]$$

$$= \sum_{k=1}^{T} \left[ \bar{V}_1^{\pi^k,k}(\bar{\mathbf{s}}_1) - V_1^{\pi^k}(\bar{\mathbf{s}}_1) \right] + T\epsilon. \tag{31}$$

The remainder of the proof focuses on bounding the first term.

### D.4.2. Recursive Decomposition of the Value Gap

For each step $h$ in episode $k$, since policy $\pi^k$ is deterministic and selects $\mathbf{a}_h^k$ at state $\mathbf{s}_h^k$, we have

$$\bar{V}_h^{\pi^k,k}(\mathbf{s}_h^k) - V_h^{\pi^k}(\mathbf{s}_h^k) = \bar{Q}_h^{\pi^k,k}(\mathbf{s}_h^k, \mathbf{a}_h^k) - Q_h^{\pi^k}(\mathbf{s}_h^k, \mathbf{a}_h^k). \tag{32}$$

Applying the Bellman equations (28) and (29), we have

$$\bar{Q}_h^{\pi^k,k}(\mathbf{s}, \mathbf{a}) = r(\mathbf{s}, \mathbf{a}) + \sum_{i=1}^{K} b(N_{i,h}^k(s^i, a^i)) + \sum_{\mathbf{s}'} \widehat{\mathbf{P}}_h^k(\mathbf{s}'|\mathbf{s}, \mathbf{a}) \bar{V}_{h+1}^{\pi^k,k}(\mathbf{s}'),$$

$$Q_h^{\pi^k}(\mathbf{s}, \mathbf{a}) = r(\mathbf{s}, \mathbf{a}) + \sum_{\mathbf{s}'} \mathbf{P}_h(\mathbf{s}'|\mathbf{s}, \mathbf{a}) V_{h+1}^{\pi^k}(\mathbf{s}'),$$

where $b(N) = H\sqrt{\frac{2S\iota}{N}} + \frac{3HS\iota}{N}$. Taking the difference at $(\mathbf{s}_h^k, \mathbf{a}_h^k)$, we can get

$$\bar{Q}_h^{\pi^k,k}(\mathbf{s}_h^k, \mathbf{a}_h^k) - Q_h^{\pi^k}(\mathbf{s}_h^k, \mathbf{a}_h^k)$$

$$= \sum_{\mathbf{s}'} \widehat{\mathbf{P}}_h^k(\mathbf{s}'|\mathbf{s}_h^k, \mathbf{a}_h^k) \bar{V}_{h+1}^{\pi^k,k}(\mathbf{s}') - \sum_{\mathbf{s}'} \mathbf{P}_h(\mathbf{s}'|\mathbf{s}_h^k, \mathbf{a}_h^k) V_{h+1}^{\pi^k}(\mathbf{s}') + \sum_{i=1}^{K} b(N_{i,h}^k(s_h^{i,k}, a_h^{i,k}))$$

$$= \sum_{\mathbf{s}'} [\widehat{\mathbf{P}}_h^k - \mathbf{P}_h](\mathbf{s}'|\mathbf{s}_h^k, \mathbf{a}_h^k) \bar{V}_{h+1}^{\pi^k,k}(\mathbf{s}') + \sum_{\mathbf{s}'} \mathbf{P}_h(\mathbf{s}'|\mathbf{s}_h^k, \mathbf{a}_h^k) [\bar{V}_{h+1}^{\pi^k,k}(\mathbf{s}') - V_{h+1}^{\pi^k}(\mathbf{s}')]$$

$$+ \sum_{i=1}^{K} b(N_{i,h}^k(s_h^{i,k}, a_h^{i,k})). \tag{33}$$

We further refine the first term by adding and subtracting $\sum_{\mathbf{s}'} [\widehat{\mathbf{P}}_h^k - \mathbf{P}_h](\mathbf{s}'|\mathbf{s}_h^k, \mathbf{a}_h^k) V_{h+1}^{\pi^k}(\mathbf{s}')$:

$$\sum_{\mathbf{s}'} [\widehat{\mathbf{P}}_h^k - \mathbf{P}_h](\mathbf{s}'|\mathbf{s}_h^k, \mathbf{a}_h^k) \bar{V}_{h+1}^{\pi^k,k}(\mathbf{s}')$$

$$= \sum_{\mathbf{s}'} [\widehat{\mathbf{P}}_h^k - \mathbf{P}_h](\mathbf{s}'|\mathbf{s}_h^k, \mathbf{a}_h^k) V_{h+1}^{\pi^k}(\mathbf{s}') + \sum_{\mathbf{s}'} [\widehat{\mathbf{P}}_h^k - \mathbf{P}_h](\mathbf{s}'|\mathbf{s}_h^k, \mathbf{a}_h^k) [\bar{V}_{h+1}^{\pi^k,k}(\mathbf{s}') - V_{h+1}^{\pi^k}(\mathbf{s}')]. \tag{34}$$

### D.4.3. Bounding the Decomposed Terms

Since $V_{h+1}^{\pi^k}(\mathbf{s}') \in [0, H]$ for all $\mathbf{s}'$, by Lemma 10:

$$\left| \sum_{\mathbf{s}'} [\widehat{\mathbf{P}}_h^k - \mathbf{P}_h](\mathbf{s}'|\mathbf{s}_h^k, \mathbf{a}_h^k) V_{h+1}^{\pi^k}(\mathbf{s}') \right| \leq \sum_{i=1}^{K} b(N_{i,h}^k(s_h^{i,k}, a_h^{i,k})). \tag{35}$$

From Lemma 11, we have $\bar{V}_{h+1}^{\pi^k,k}(\mathbf{s}') - V_{h+1}^{\pi^k}(\mathbf{s}') \geq 0$ for all $\mathbf{s}'$. Applying Lemma 9:

$$\sum_{\mathbf{s}'}[\widehat{\mathbf{P}}_h^k - \mathbf{P}_h](\mathbf{s}'|\mathbf{s}_h^k, \mathbf{a}_h^k)[\bar{V}_{h+1}^{\pi^k,k}(\mathbf{s}') - V_{h+1}^{\pi^k}(\mathbf{s}')]$$

$$\leq \frac{1}{H}\sum_{\mathbf{s}'}\mathbf{P}_h(\mathbf{s}'|\mathbf{s}_h^k, \mathbf{a}_h^k)[\bar{V}_{h+1}^{\pi^k,k}(\mathbf{s}') - V_{h+1}^{\pi^k}(\mathbf{s}')] + \sum_{i=1}^{K}\frac{10SH^2K\iota}{N_{i,h}^k(s_h^{i,k}, a_h^{i,k})}. \tag{36}$$

Combining (35), (36), (33), and (34):

$$\bar{V}_h^{\pi^k,k}(\mathbf{s}_h^k) - V_h^{\pi^k}(\mathbf{s}_h^k) \leq \left(1 + \frac{1}{H}\right)\sum_{\mathbf{s}'}\mathbf{P}_h(\mathbf{s}'|\mathbf{s}_h^k, \mathbf{a}_h^k)[\bar{V}_{h+1}^{\pi^k,k}(\mathbf{s}') - V_{h+1}^{\pi^k}(\mathbf{s}')]$$

$$+ 2\sum_{i=1}^{K}b(N_{i,h}^k(s_h^{i,k}, a_h^{i,k})) + \sum_{i=1}^{K}\frac{10SH^2K\iota}{N_{i,h}^k(s_h^{i,k}, a_h^{i,k})}. \tag{37}$$

### D.4.4. INTRODUCTION OF MARTINGALE STRUCTURE

Define the difference term $\xi_h^k$ as

$$\xi_h^k := \sum_{\mathbf{s}'}\mathbf{P}_h(\mathbf{s}'|\mathbf{s}_h^k, \mathbf{a}_h^k)[\bar{V}_{h+1}^{\pi^k,k}(\mathbf{s}') - V_{h+1}^{\pi^k}(\mathbf{s}')] - [\bar{V}_{h+1}^{\pi^k,k}(\bar{\mathbf{s}}_{h+1}^k) - V_{h+1}^{\pi^k}(\mathbf{s}_{h+1}^k)],$$

and the deterministic error term:

$$\eta_h^k := 2\sum_{i=1}^{K}b(N_{i,h}^k(s_h^{i,k}, a_h^{i,k})) + \sum_{i=1}^{K}\frac{10SH^2K\iota}{N_{i,h}^k(s_h^{i,k}, a_h^{i,k})}.$$

With these definitions, (37) becomes:

$$\bar{V}_h^{\pi^k,k}(\mathbf{s}_h^k) - V_h^{\pi^k}(\mathbf{s}_h^k) \leq \left(1 + \frac{1}{H}\right)[\bar{V}_{h+1}^{\pi^k,k}(\bar{\mathbf{s}}_{h+1}^k) - V_{h+1}^{\pi^k}(\mathbf{s}_{h+1}^k)] + \left(1 + \frac{1}{H}\right)\xi_h^k + \eta_h^k. \tag{38}$$

### D.4.5. RECURSIVE EXPANSION

Applying (38) recursively from $h = 1$ to $h = H$:

$$\bar{V}_1^{\pi^k,k}(\bar{\mathbf{s}}_1^k) - V_1^{\pi^k}(\bar{\mathbf{s}}_1^k)$$

$$\leq \left(1 + \frac{1}{H}\right)^H[\bar{V}_{H+1}^{\pi^k,k}(\bar{\mathbf{s}}_{H+1}^k) - V_{H+1}^{\pi^k}(\mathbf{s}_{H+1}^k)] + \sum_{h=1}^{H}\left(1 + \frac{1}{H}\right)^{H-h+1}\xi_h^k + \sum_{h=1}^{H}\left(1 + \frac{1}{H}\right)^{H-h}\eta_h^k. \tag{39}$$

Since $\bar{V}_{H+1}^{\pi^k,k}(\bar{\mathbf{s}}_{H+1}^k) = V_{H+1}^{\pi^k}(\mathbf{s}_{H+1}^k) = 0$ and $(1 + \frac{1}{H})^H \leq e$:

$$\bar{V}_1^{\pi^k,k}(\bar{\mathbf{s}}_1^k) - V_1^{\pi^k}(\bar{\mathbf{s}}_1^k) \leq e\sum_{h=1}^{H}\xi_h^k + e\sum_{h=1}^{H}\eta_h^k. \tag{40}$$

Summing over all $T$ episodes, we can get

$$\mathcal{R}_{T,\frac{1}{2}} \leq \sum_{k=1}^{T}[\bar{V}_1^{\pi^k,k}(\bar{\mathbf{s}}_1) - V_1^{\pi^k}(\bar{\mathbf{s}}_1)] + T\epsilon$$

$$\leq e\sum_{k=1}^{T}\sum_{h=1}^{H}\xi_h^k + e\sum_{k=1}^{T}\sum_{h=1}^{H}\eta_h^k + T\epsilon. \tag{41}$$

### D.4.6. BOUNDING THE DETERMINISTIC EXPLORATION COST

Expanding the definition of $\eta_h^k$ with $b(N) = H\sqrt{\frac{2S\iota}{N}} + \frac{3HS\iota}{N}$:

$$\sum_{k=1}^{T}\sum_{h=1}^{H}\eta_h^k \leq \sum_{k=1}^{T}\sum_{h=1}^{H}\left(\sum_{i=1}^{K}\frac{16SH^2K\iota}{N_{i,h}^k(s_h^{i,k}, a_h^{i,k})} + 2H\sum_{i=1}^{K}\sqrt{\frac{2S\iota}{N_{i,h}^k(s_h^{i,k}, a_h^{i,k})}}\right),$$

where we used $10SH^2K\iota + 2 \cdot 3HS\iota \leq 16SH^2K\iota$.

For the first term, reordering summations and grouping by state-action pairs:

$$\sum_{k=1}^{T}\sum_{h=1}^{H}\sum_{i=1}^{K}\frac{16SH^2K\iota}{N_{i,h}^k(s_h^{i,k}, a_h^{i,k})} = 16SH^2K\iota\sum_{i=1}^{K}\sum_{h=1}^{H}\sum_{(s,a)}\sum_{\ell=1}^{N_{i,h}^T(s,a)}\frac{1}{\ell}$$

$$\leq 16SH^2K\iota\sum_{i=1}^{K}\sum_{h=1}^{H}\sum_{(s,a)}(1 + \log N_{i,h}^T(s,a)) \leq 32S^2AH^3K^2\iota\log T,$$

where we used $\sum_{\ell=1}^{n}\frac{1}{\ell} \leq 1 + \log n$ and $N_{i,h}^T(s,a) \leq T$.

For the second term, using $\sum_{\ell=1}^{n}\frac{1}{\sqrt{\ell}} \leq 1 + 2\sqrt{n}$:

$$2H\sum_{k=1}^{T}\sum_{h=1}^{H}\sum_{i=1}^{K}\sqrt{\frac{2S\iota}{N_{i,h}^k(s_h^{i,k}, a_h^{i,k})}} = 2H\sqrt{2S\iota}\sum_{i=1}^{K}\sum_{h=1}^{H}\sum_{(s,a)}\sum_{\ell=1}^{N_{i,h}^T(s,a)}\frac{1}{\sqrt{\ell}}$$

$$\leq 2H\sqrt{2S\iota}\sum_{i=1}^{K}\sum_{h=1}^{H}\sum_{(s,a)}(1 + 2\sqrt{N_{i,h}^T(s,a)})$$

$$\leq 2\sqrt{2S\iota}H^2KSA + 2H\sqrt{2S\iota}\sum_{i=1}^{K}\sum_{h=1}^{H}\sum_{(s,a)}2\sqrt{N_{i,h}^T(s,a)}$$

$$\leq 2\sqrt{2S\iota}H^2KSA + 4H^2KS\sqrt{2AT\iota}.$$

Combining:

$$\sum_{k=1}^{T}\sum_{h=1}^{H}\eta_h^k \leq 32S^2AH^3K^2\iota\log T + 4H^2KS\sqrt{2AT\iota} + 2\sqrt{2S\iota}H^2KSA. \tag{42}$$

### D.4.7. BOUNDING THE MARTINGALE TERMS

Let $\mathcal{F}_h^k$ denote the $\sigma$-algebra generated by all randomness up to step $h$ of episode $k$. The value gap $\bar{V}_{h+1}^{\pi^k,k}(\mathbf{s}) - V_{h+1}^{\pi^k}(\mathbf{s})$ is $\mathcal{F}_h^k$-measurable for any $\mathbf{s}$ (since it depends only on data through episode $k-1$), while $\mathbf{s}_{h+1}^k$ is $\mathcal{F}_{h+1}^k$-measurable but not $\mathcal{F}_h^k$-measurable.

Computing the conditional expectation:

$$\mathbb{E}[\xi_h^k|\mathcal{F}_h^k] = \sum_{\mathbf{s}'}\mathbf{P}_h(\mathbf{s}'|\mathbf{s}_h^k, \mathbf{a}_h^k)[\bar{V}_{h+1}^{\pi^k,k}(\mathbf{s}') - V_{h+1}^{\pi^k}(\mathbf{s}')] - \mathbb{E}\left[\bar{V}_{h+1}^{\pi^k,k}(\bar{\mathbf{s}}_{h+1}^k) - V_{h+1}^{\pi^k}(\mathbf{s}_{h+1}^k)\Big|\mathcal{F}_h^k\right].$$

Since $\mathbf{s}_{h+1}^k \sim \mathbf{P}_h(\cdot|\mathbf{s}_h^k, \mathbf{a}_h^k)$ given $\mathcal{F}_h^k$:

$$\mathbb{E}\left[\bar{V}_{h+1}^{\pi^k,k}(\bar{\mathbf{s}}_{h+1}^k) - V_{h+1}^{\pi^k}(\mathbf{s}_{h+1}^k)\Big|\mathcal{F}_h^k\right] = \sum_{\mathbf{s}'}\mathbf{P}_h(\mathbf{s}'|\mathbf{s}_h^k, \mathbf{a}_h^k)[\bar{V}_{h+1}^{\pi^k,k}(\mathbf{s}') - V_{h+1}^{\pi^k}(\mathbf{s}')].$$

Therefore:

$$\mathbb{E}[\xi_h^k | \mathcal{F}_h^k] = 0, \tag{43}$$

confirming that $\{\xi_h^k\}$ is a martingale difference sequence.

Since $0 \le \bar{V}_{h+1}^{\pi^k,k}(\mathbf{s}'), V_{h+1}^{\pi^k}(\mathbf{s}') \le H$:

$$|\xi_h^k| \le \sum_{\mathbf{s}'} \mathbf{P}_h(\mathbf{s}'|\mathbf{s}_h^k, \mathbf{a}_h^k)|\bar{V}_{h+1}^{\pi^k,k}(\mathbf{s}') - V_{h+1}^{\pi^k}(\mathbf{s}')| + |\bar{V}_{h+1}^{\pi^k,k}(\bar{\mathbf{s}}_{h+1}^k) - V_{h+1}^{\pi^k}(\mathbf{s}_{h+1}^k)| \le 4H.$$

Applying Azuma-Hoeffding with $TH$ martingale differences each bounded by $4H$, with probability at least $1 - \frac{\delta}{3}$:

$$\sum_{k=1}^{T}\sum_{h=1}^{H} \xi_h^k \le \sqrt{2TH \cdot (4H)^2 \cdot \log(3/\delta)} = 4H\sqrt{2TH\log(3/\delta)}.$$

Since $\iota = \log(10S^2AT/\delta) \ge \log(3/\delta)$:

$$\sum_{k=1}^{T}\sum_{h=1}^{H} \xi_h^k \le 4H\sqrt{2TH\iota}. \tag{44}$$

### D.4.8. FINAL ASSEMBLY

Combining (41), (42), and (44), and applying the union bound on the event $\mathcal{E}$ in Lemma 6 and the Azuma's inequality used to bound $\sum_{k=1}^{T}\sum_{h=1}^{H} \xi_h^k$, we have that with probability at least $1 - \delta$:

$$\mathcal{R}_{T,\frac{1}{2}} \le 4eH\sqrt{2TH\iota} + e(32S^2AH^3K^2\iota\log T + 4H^2KS\sqrt{2AT\iota} + 2\sqrt{2S\iota}H^2KSA) + T\epsilon$$
$$\le 4eH\sqrt{2TH\iota} + 96S^2AH^3K^2\iota\log T + 4eH^2KS\sqrt{2AT\iota} + T\epsilon.$$

Choosing $\epsilon = O(1/T)$ and absorbing constants into $O$ notation:

$$\mathcal{R}_{T,\frac{1}{2}} = O\left(S^2AH^3K^2\iota\log T + H^2KS\sqrt{AT\iota}\right).$$

Since $\iota := \log(6S^2ATHK/\delta)$, we can get

$$\mathcal{R}_{T,\frac{1}{2}} = O\left(S^2AH^3K^2\log(SATHK/\delta)\log T + H^2KS\sqrt{AT\log(SATHK/\delta)}\right).$$

This establishes the desired $O(\sqrt{T})$ regret rate, completing the proof.

## E. Technical Lemmas

**Lemma 12** (Bernstein's Inequality). *Let $x_1, ..., x_n$ be independent bounded random variables such that $\mathbb{E}[x_i] = 0$ and $|x_i| \le \xi$ with probability 1. Let $\sigma^2 = \frac{1}{n}\sum_{i=1}^{n} \mathrm{Var}[x_i]$, then with probability $1 - \delta$ we have*

$$\frac{1}{n}\sum_{i=1}^{n} x_i \le \sqrt{\frac{2\sigma^2 \cdot \log(1/\delta)}{n}} + \frac{2\xi}{3n}\log(1/\delta)$$

**Lemma 13** (Hoeffding's Inequality). *Let $X_1, ..., X_N$ be independent random variables such that $X_i$ is $R$-sub-Gaussian and $\mathbb{E}[X_i] = \mu$ for all $i$. Let $\overline{X} = \frac{1}{N}\sum_{i=1}^{N} X_i$. Then for any $t > 0$,*

$$P(|\overline{X} - \mu| \ge t) \le 2\exp\{-\frac{Nt^2}{2R^2}\}.$$

