# OpenReview forum: "Multi-Agent Reinforcement Learning with Submodular Reward"
_ICML.cc/2026/Conference — ICML 2026 regular_

### Official Review · Reviewer_wApX · 2026-02-27

**Soundness:** 2
**Presentation:** 2
**Significance:** 2
**Originality:** 2
**Overall Recommendation:** 3
**Confidence:** 5

**Summary:**

The paper addresses deep MARL for multi-agent Markov decision processes. It leverages submodularity present in the reward models of many real-world applications. The authors use this setting to study the sample efficiency of two MARL algorithms they introduced. The takeaways include a proof that greedy policy optimisation achieves a 1/2-approximation with polynomial complexity in the number of agents and two algorithms addressing model-based and model-free settings.

**Compliance With Llm Reviewing Policy:**

Affirmed.

**Key Questions For Authors:**

- Position the contribution with respect to existing similar structural assumptions
- Sequentialisation is now well formalised in general MARL;
what does submodularity bring that we could not get in the general MARL?

**Limitations:**

Yes.

**Strengths And Weaknesses:**

- Soundness.

The paper is mainly theoretical -- no experiments are provided.

The assumptions on the state space seem quite strong: (i) agents share the same world with the same state space; and (ii) yet they have independent dynamics. This is a very special setting, and I could not find a non-trivial example that meet these requirements. The authors use multi-drone surveillance and collaborative exploration as motivating examples. But, I was unable to relate these assumptions to those applications. Either the state space is partitioned, in which case each agent has its own state space, or the state space is common, then no agent can independently move and yet ensure collision-free motions.

The assumptions on the policies should be better motivated and justified. Policies in cooperative settings need not be randomised. Since the paper focuses on tabular approaches I was wondering why we consider stochastic policies instead of deterministic ones, which can reinforce coordination much better than randomised ones. A careful look at the algorithms revealed indeed that only deterministic policies are computed in practice.

The proofs are standard and seem correct.


- Presentation.

The paper could do a better job positioning the work with respect to related work on Network distributed Markov decision processes [0,1] and sequential cooperative reformulations of Multi-agent (Dec-)MDPs [2,3]. Papers [0,1] are closely related with the present work: transition independent, reward weakly coupled, although states are factored. The structure of the optimal value function is shown to be factored. We expect a similar result here, even without any approximation. It is not clear how do these papers differ from the present work. In [0,1], the complexity of the greedy policy operator is exponential with the size of the largest cluster of agents, i.e., number of agents sharing a joint reward function. From what I understand, exploiting submodularity enable the complexity to drop from exponential in the worst case to polynomial when we focus on 1/2-approximation. The idea is a sequentialisation of greedy policy optimisation, that also appears in [2,3].


Reference.
[0] Kumar, A.,  Zilberstein, S. (2009, May). Constraint-Based Dynamic Programming for Decentralized POMDPs with Structured Interactions. In Proceedings of the 8th International Conference on Autonomous Agents and Multiagent Systems (AAMAS 2009).

[1] Dibangoye, J., Amato, C., Buffet, O., & Charpillet, F. (2014). Exploiting separability in multiagent planning with continuous-state MDPs. In AAMAS 2014-13th International Conference on Autonomous Agents and Multiagent Systems. ACM.

[2] Peralez, J., Delage, A., Castellini, J., Cunha, R. F.,  Dibangoye, J. S. (2025). Optimally solving simultaneous-move dec-POMDPs: The sequential central planning approach. In Proceedings of the AAAI Conference on Artificial Intelligence (Vol. 39, No. 22, pp. 23276-23285).

[3] Bertsekas, D. P. (2022). Abstract Dynamic Programming. 3rd ed., Athena Scientific.


- Significance.

I think the paper could become an important contribution to the field of MARL providing clarifications on the settings and the connexion with closely related work is made clear. Currently, the key ingredient for the analysis seems to be the ability to sequentialise the greedy policy optimisation. Similar results have been established in the general MARL settings, eg [2,3]. What does the subclass of submodular reward models provide that cannot be obtained in general sequentialised approaches?

---

> ### Author Rebuttal · Authors · 2026-03-31
>
> >1. The assumptions on the state space seem quite strong ...
>
> Thank you for raising this important point. We would like to clarify the modeling assumptions and their intended scope.
>
> In our formulation, agents operate in a shared environment, meaning they interact with a common global state space. The assumption of independent dynamics refers to the transition structure: each agent’s state evolution depends only on its own action (and possibly exogenous randomness), rather than directly on the actions of other agents. This type of decoupling is commonly adopted in large-scale multi-agent systems to enable tractable analysis.
>
> We agree that collision avoidance and other safety constraints introduce coupling between agents and are important in many real-world settings. Our current framework abstracts away these constraints to focus on the core learning problem. In practice, such constraints can be incorporated in several standard ways, e.g., by augmenting the state space, adding penalty terms to the reward, or introducing safety filters. Extending our theoretical framework to explicitly handle coupled constraints is an interesting direction for future work, and we will clarify this limitation in the revision.
>
> >2. ...I was wondering why we consider stochastic policies instead of deterministic ones...
>
> We thank the reviewer for this observation. Indeed, the policies produced by our method are deterministic. This, in fact, highlights the generality of our results: although our algorithm outputs deterministic policies, the performance guarantees hold with respect to a broader class of policies, including randomized ones (e.g., $\epsilon$-greedy or other stochastic sampling-based strategies).
>  This demonstrates that our method remains competitive even when compared against a strictly more general class of policies beyond deterministic ones.
>
> In particular, we establish that for any  **randomized, non-factorized** policy, our method outputs a policy whose value is guaranteed to achieve at least a constant-factor approximation of $1/2$ of that policy.
>
>
> >3. The paper could do a better job positioning the work with respect to related work on Network distributed Markov decision processes [0,1] and sequential cooperative reformulations of Multi-agent (Dec-)MDPs [2,3]. ...
>
> Thank you for pointing out these related papers!
>
>
> 1. **Comparison with [0,1]**: Although both our work and the mentioned papers assumed weakly coupled policies, our formulation is **more general in terms of the reward structure**. Papers [0,1] assumes the reward can be factored into the additive sum of groups of subagents. This additive function can in fact be viewed as a special case of submodular function. Therefore, our setting is more general. In fact, the MARL setting with submodular rewards captures a range of important applications where additive assumptions fail. For example, in map-exploration where the reward is modeled as entropy, the additive assumption doesn't work if different agents have overlap in their view. Similarly, in multi-robot task allocation, redundant execution of the same task does not yield additive rewards. These scenarios are naturally modeled by submodular objectives but fall outside the scope of additive formulations.
>
>
>
> 2.  **Comparison with [2,3]**:  [2,3] discuss sequential multi-agent reinforcement learning. However, [3] discuss the sequential multi-agent reinforcement learning under the zero-sum setting, which is the competitive MARL setting that is different from us. [2] discuss the sequential optimization in coorperative MARL. Although the sequential reformulation is theoretically exact, the resulting policy depends on the state space of the agents history, which can be large. To deal with this, the proposed oSARSA algorithm does not solve the resulting dynamic program exactly. Instead, it relies on approximate value function representations and trajectory sampling, which introduces bias in the evaluation of future rewards. Consequently, the resulting policy doesn't have theoretical guarantee. Our method, on the other hand, directly optimizes the objective with provable guarantees, thereby achieving significantly better performance.
>
>
> > I think the paper could become an important contribution to the field of MARL providing clarifications on the settings...
>
> We would like to thank the reviewer for the recognition of the importantce and contribution of our work. The comparison with [0,1,2,3] has been presented in the question above.

---

> > ### Author Rebuttal · Reviewer_wApX · 2026-03-31
> >
> > Thank you for the responses to some of my questions or criticisms. I am a bit concerned about the depth and the breadth of the responses, though.
> >
> > 1. Regarding assumptions. The authors suggest that it is common to make such strong assumptions for theoretical papers and that it is possible to recover the full generality of the model via the reward model. Well, this argument needs either a reference with a formal proof of the latter claim and references for the former claim.
> >
> > 2. In Dec-(PO)MDPs, the optimal policies are deterministic. Searching in the randomised policy space needs to be justified. In general, it is harder to prove that a property that holds for randomised policies also holds in deterministic ones. But the opposite is often obtained for free—so I will not make it an advantage of the present paper.
> >
> > 3. The analysis of the related work the authors provided is not satisfactory. I will not claim that the present paper is "more general in terms of the reward structure" than [1]. The reason is simple: whenever we only have one cluster of agents, this model boils down to a Dec-POMDP, which is the most general discrete cooperative model.
> >
> > 4. Regarding sequential approaches, I was expecting the authors to compare the theories, which are exact, and reflect on how these theories could inspire their own contributions. BTW, oSarsa is actually guaranteed to converge asymptotically to the optimal solution under the right hyperparameters.

---

> > > ### Author Response · Authors · 2026-04-05
> > >
> > > We thank the reviewer for the thoughtful follow-up. Below, we address each concern respectively:
> > >
> > > 1. We agree that our assumptions are strong and should be justified further. The assumptions allow us to study a tractable class of cooperative MARL problems where (1) the curse of dimensionality can be overcome, yielding algorithms with polynomial dependence on $K$ and (2) submodular rewards capture richer agent interactions than linear structures permit. Similar transition-independence assumptions appear in prior work on decentralized MDPs and weakly coupled systems [0, 1], where they serve the same purpose. We also clarify that we do not claim to recover the full generality of Dec-(PO)MDPs — our framework targets a structured, tractable subclass where agents interact through the reward while maintaining decoupled dynamics.
> > >
> > > 2. We agree that optimal policies in Dec-(PO)MDPs can be taken to be deterministic. **Our algorithm is also deterministic, as it follows a sequential greedy construction.**
> > >
> > >     We also agree that limiting the search space to the deterministic space and applying the Bellman equation can reduce the computation complexity. We can reduce the **time complexity of evaluating a deterministic policy** from $O(S^{2K}A^{K})$ to $O(S^{2K})$ for each step. However, this **is still exponential in terms of the number of agents**. The key idea in our approach is that by exploiting submodularity, and by proposing the sequential greedy approach, we can overcome this curse of dimensionality from exponential in $K$ to polynomial in $K$.
> > >
> > >     Our intention in discussing randomized policies was not to claim an advantage over deterministic policies, but to emphasize that our guarantees hold **without requiring restrictive structural assumptions on the policy class** (e.g., decomposable or deterministic across agents).
> > >
> > > 3. We agree that the reward modeling in [1] is more general within each cluster of agents. In particular, the two frameworks differ in:
> > >
> > > - **Within-cluster generality.** In [1], the reward within each cluster is unrestricted, making the model more expressive at the intra-cluster level. In particular, when there is only one cluster, the model subsumes a full Dec-POMDP.
> > >
> > > - **Cross-group reward structure.** In [1], the global reward decomposes *additively* across clusters:
> > >   $$f(A_{[K]}) = \sum_{i=1}^N f(A_{U_i}),$$ where $\{U_i\}$ denotes a partition of agents. This cross-cluster additive structure of [1] can be recovered as a special case of submodularity. Specifically, by submodularity we have $f(A_{U_i}) \geq f(\bigcup_{m' \in [i]} A_{U_{m'}}) - f(\bigcup_{m' \in [i-1]} A_{U_{m'}})$ for any $i$. When the submodularity inequality holds with equality across groups — a strictly stronger condition than submodularity alone — telescoping then recovers the additive assumption in [1].
> > >
> > >     In summary, these two frameworks operate at different levels of granularity — cross-cluster additivity vs. cross-agent submodularity — and are not directly comparable in full generality. [1] is more general within clusters, while our formulation is more general in modeling reward interactions across clusters of agents.
> > >
> > >     Finally, we highlight that submodularity introduces challenges absent under additive rewards — coupled marginal rewards and the failure of sequential optimization to recover the optimal joint policy — requiring fundamentally new techniques.
> > >
> > > 4. We appreciate the comment and will include a discussion of the sequential approach in Dec-POMDP. Our contribution differs from [2] in the following aspects:
> > >
> > >     - **Approximation guarantee against the optimal joint policy:** In Dec-POMDPs, the algorithms (such as oSARSA) aim to find the optimal *decentralized* policy. In contrast,  our $1/2$-approximation ratio (Theorem 1) is established against the optimal *joint* policy. The policy our algorithm outputs admits decentralized execution, but this is a result of our methodology rather than formulation.
> > >
> > >     - **Different form of sequentiality:** In [2], the sequential optimization interleaves across agents and time steps: at each simultaneous time step $t$, agents' decision rules are selected one-by-one, then advances to $t+1$ and repeats. In our algorithm, the optimization is sequential at the agent level: we solve for agent 1's complete policy across all time steps, fix it, then solve for agent 2, and so on.
> > >
> > >     - **Backup complexity independent of planning horizon:** The previous structural difference leads to a concrete complexity advantage. In [2], the SOCs summarize the history of actions and observations, whose size can grows exponentially in the decision epoch $\tau$. In our setting, each agent's policy depends only on its *own current state*, as all inter-agent coordination is handled during training via the marginal reward estimation. The per-agent policy size is $O(|S||A|H)$, and the backup complexity has no dependence on horizon beyond the linear factor from iterating over time steps.

---

### Official Review · Reviewer_Mmtm · 2026-03-09

**Soundness:** 3
**Presentation:** 3
**Significance:** 4
**Originality:** 4
**Overall Recommendation:** 5
**Confidence:** 2

**Summary:**

Overall, the authors assess a general context of cooperative MARL where the joint reward function exhibits submodularity. The authors intend to investigate a notable area where agents' contributions overlap, leading to diminishing marginal returns. The paper formalizes the MAMDP-SR framework and proposes a marginal value decomposition method to overcome the curse of dimensionality. For known dynamics, they introduce a Greedy Policy Optimization algorithm achieving a 1/2-approximation. For unknown dynamics, they propose UCB-GVI, achieving a sublinear 1/2-regret bound of $\mathcal{O}(H^2 KS\sqrt{AT})$.

The paper tackles a novel and practically relevant problem with highly rigorous and impressive theoretical contributions. However, the complete absence of empirical evaluation and the reliance on the independent transition assumption slightly limit its immediate impact.

**Compliance With Llm Reviewing Policy:**

Affirmed.

**Final Justification:**

The paper is well writen, and the idea of RL with submodular reward inspires me. I would like to raise my score.

**Key Questions For Authors:**

1. Could the authors extend the theoretical framework and regret analysis to settings where transition dynamics are coupled or dependent on joint actions?

2. Are there any plans to include empirical experiments to demonstrate that UCB-GVI learns effectively in practice?

3. Have the authors considered any methods to bypass or reduce the heavy trajectory sampling required for marginal reward estimation?

**Limitations:**

Yes

**Strengths And Weaknesses:**

Strong points:
1. Strong Motivation: Shifting from additive to submodular rewards is highly relevant for real-world MARL applications like coverage and exploration.

2. Solid Theory: The paper provides rigorous theoretical guarantees, including the first sublinear regret bound for MARL with submodular rewards.

3. Tractability: The proposed algorithms successfully reduce the exponential complexity of joint policy optimization to polynomial time/space.

Weak points:
1. No Empirical Evaluation: The paper is entirely theoretical. The lack of experiments makes it hard to assess practical viability and sample efficiency.

2. High Sampling Overhead: The trajectory sampling required to estimate marginal rewards introduces a massive computational burden in practice.

---

> ### Author Rebuttal · Authors · 2026-03-31
>
> ### **Response to Additional Questions**
>
> We thank the reviewer for these thoughtful and forward-looking questions. We address them below.
>
> ---
>
> **Q1. Could the authors extend the theoretical framework and regret analysis to settings where transition dynamics are coupled or depend on joint actions?**
>
> This is an important and interesting direction. Our current analysis focuses on settings with decoupled transition dynamics, which allows us to exploit structure for both tractability and sharp regret guarantees. Extending the framework to coupled dynamics where transitions depend on joint actions introduces significant technical challenges, as it breaks the separability used in both the estimation and analysis.
>
> However, we believe our approach provides a useful framework for enabling the decomposition and separation of the reward function that exhibits submodularity, and thus provides a foundation for such extensions. In particular, incorporating joint-action-dependent transitions would likely require new concentration tools and a different decomposition of the value function. We view this as an exciting direction for future work and will clarify these limitations and potential extensions in the final version.
>
> ---
>
> **Q2. Are there any plans to include empirical experiments to demonstrate that UCB-GVI learns effectively in practice?**
>
> Thank you for the question. We acknowledge that the current submission primarily focuses on the theoretical contributions and regret guarantees. This emphasis is consistent with a substantial body of work in ICML that studies reinforcement learning from a theoretical perspective [1,2,3]. That said, we agree that empirical validation is important. We will implement UCB-GVI and evaluate it on standard benchmarks in future work.
>
> - [1]. Qiao, Dan, et al. "Sample-efficient reinforcement learning with loglog (t) switching cost." International Conference on Machine Learning. PMLR, 2022.
> - [2]. Jin, Chi, et al. "Reward-free exploration for reinforcement learning." International Conference on Machine Learning. PMLR, 2020.
> - [3]. Yang, Lin, and Mengdi Wang. "Sample-optimal parametric q-learning using linearly additive features." International conference on machine learning. PMLR, 2019.
>
> ---
>
> **Q3. Have the authors considered methods to bypass or reduce the heavy trajectory sampling required for marginal reward estimation?**
>
> Thank you for raising this point. Indeed, the current approach relies on trajectory sampling to estimate marginal rewards. Although this can convert the exponential computational complexity into polynomial ones, this can be computationally intensive in practice.
>
> There are several promising directions to mitigate this:
> - **Variance reduction techniques** (e.g., importance sampling or control variates),
> - **Model-based estimation**, when approximate dynamics are available,
> - **Function approximation or parametric models** to generalize marginal contributions across states, such as in the case of linear function approximation.
>
> Exploring these alternatives could significantly improve sample efficiency and scalability. While a full treatment is beyond the scope of the current paper, we view this as an important avenue for future work and will discuss it more explicitly in the revision.

---

> > ### Author Rebuttal · Reviewer_Mmtm · 2026-04-04
> >
> > thanks for the response. I support acceptance of this paper.

---

### Official Review · Reviewer_xJUJ · 2026-03-11

**Soundness:** 3
**Presentation:** 4
**Significance:** 2
**Originality:** 3
**Overall Recommendation:** 4
**Confidence:** 3

**Summary:**

This paper studies cooperative multi-agent RL with submodular rewards. They provide a formal theoretical framework and propose a upper confidence bound (UCB) algorithm. The algorithm utilizes the submodularity to circumvent the curse of dimensionality and achieve a 1/2 approximation of the optimal value function (analogous to the 1/2 approximation for greedy solutions monotone submodular approximation). They also provide an analysis of the regret of the algorithm (up to the 1/2 approximation).

**Compliance With Llm Reviewing Policy:**

Affirmed.

**Final Justification:**

Overall, I keep my current score and the authors have addressed my concerns. If the authors can strengthen the applications section as indicated in their rebuttal, I think it would reinforce the result better.

**Key Questions For Authors:**

1) What specific baselines, problems, and applications do you anticipate your framework to be used for? How would you model them concretely with your framework (e.g. partial observability from surveillance applications)?

2) What other prior works have there been that use submodularity in a MARL context, and how does it relate to your work?

3) What is the impact of the order of the agents evaluated in Algorithm 1? Can this influence the empirical performance?

**Limitations:**

Yes.

**Strengths And Weaknesses:**

Soundness 3 (Good): I did not find any significant mathematical mistakes in my reading. Overall, the paper does a good job of supporting their points. However I think this paper could benefit from introducing a more concrete motivation for the application of submodular functions to MARL. At the moment, high level motivations of "surveillance" or "tracking" are provided but a concrete applications section which formally expresses 2-3 motivating applications in the provided framework would ground the motivation for this paper more strongly. Currently, the connection is only provided at an intuitive level and is a bit vague.

Presentation 4 (Excellent): Overall, this paper was easy to understand and well written. I was new to some concepts in this work but I was able to follow.

Significance 2 (Fair): Generally I think this is a clever approach to these types of problems and is beneficial for the literature to have. Using submodular rewards in multi-agent MDP settings appear to have been done before, but not to this level of rigor [1][2]. The only hesitance I have is with the loose motivation I described in the "Soundness" section which does not entirely convince me that real life applications with submodular rewards can and should be modeled in this way (surveillance and tracking are inherently partially observable and here we are modeling with an MDP for example).

Originality 3 (Good):
I think this area is quite underexplored and this style of UCB algorithm for submodular rewards does forward the understanding in this area.

[1] Anand, Aditi, Suman Banerjee, and Dildar Ali. "Scalable submodular policy optimization via pruned submodularity graph." arXiv preprint arXiv:2507.13834 (2025).
[2] Prajapat, Manish, et al. "Submodular reinforcement learning." arXiv preprint arXiv:2307.13372 (2023).

---

> ### Author Rebuttal · Authors · 2026-03-31
>
> 1. >Weakness: However I think this paper could benefit from introducing a more concrete motivation for the application of submodular functions to MARL. At the moment, high level motivations of "surveillance" or "tracking" are provided but a concrete applications section which formally expresses 2-3 motivating applications in the provided framework would ground the motivation for this paper more strongly.   >Question: *What specific baselines, problems, and applications do you anticipate your framework to be used for? How would you model them concretely with your framework (e.g. partial observability from surveillance applications)?*
>
>     We thank the reviewer for this important question. The MARL-SR framework naturally models a broad class of cooperative multi-agent problems. We illustrate this with two applications, each specified $S, r,$ and $P_{i,h}$ in the MARL-SR setting.
>
>    **Application 1: Exploration and mapping.** Consider $K$ mobile robots equipped with range sensors (e.g., LIDAR) exploring an unknown environment to build an occupancy grid map. At each time step, each robot takes a scan from its current position, and the team's goal is to maximize the information gained about the environment.
>
>     - $S = \mathbb{R}^2$: the robot state space.
>     - $r_h(\mathbf{s}_h, \mathbf{a}_h) = I(O_1(s_1), \ldots, O_K(s_K); \mathcal{E} \mid \mathrm{grid}_h)$: the conditional mutual information between the robots' joint observations at step $h$ and the unknown environment $\mathcal{E}$, given the current grid belief. This is submodular over the set of robots' observations because overlapping sensor footprints yield diminishing informational returns.
>
>     - $P_{i,h}$: robot motion dynamics.
>
>     **Application 2: Multi-robot task allocation in warehouse logistics.** Consider $K$ mobile robots in a warehouse with $M$ spatially distributed picking tasks. Each robot can fulfill tasks within a proximity radius of its current position. The team's goal is to maximize the number of distinct tasks covered.
>
>     - $S$: robot position on the warehouse graph.
>     - $r_h(\mathbf{s}_h, \mathbf{a}_h) = | \mathrm{tasks}(s_1) \cup \cdots \cup \mathrm{tasks}(s_K) |$:  the number of distinct tasks reachable by at least one robot, where $\mathrm{tasks}(s_i) \subseteq [M]$ returns the set of tasks within robot $i$'s service radius. This is a monotone submodular set-cover function.
>     - $P_{i,h}$: robot navigation dynamics on the warehouse graph.
>
>     We acknowledge that partial observability can be an important consideration. However, we note that partial observability is not required for our framework. For instance, in surveillance settings, the reward can depend solely on whether targets are observed (e.g., coverage), without requiring agents to explicitly estimate hidden states. In such cases, the problem can be naturally modeled within our framework using observable coverage-based rewards, which are often submodular.
> 1. *What other prior works have there been that use submodularity in a MARL context, and how does it relate to your work?*
>
> Thank you for this question. Submodularity has been implicitly explored in prior multi-agent reinforcement learning works, particularly in the context of promoting diversity and exploration (e.g., [1,2]). However, these works do not explicitly model the reward function as submodular, nor do they leverage submodularity to derive theoretical guarantees.
>
> In contrast, our work explicitly formulates the reward structure through a submodular lens and uses this property to design the algorithm and establish provable guarantees. In particular, our approach enables us to derive a constant-factor approximation and regret bounds, which are not provided in prior works. This highlights a key distinction: while earlier works may exhibit submodular-like behavior empirically, our framework explicitly exploits submodularity at both the modeling and theoretical levels.
>
> [1] Controlling Behavioral Diversity in Multi-Agent Reinforcement Learning
>
> [2] The Emergence of Individuality
>
> 1. What is the impact of the order of the agents evaluated in Algorithm 1? Can this influence the empirical performance?
>
> Thank you for the insightful question. The ordering of agents in the sequential optimization does not affect the theoretical guarantees, including the $1/2$ approximation ratio and regret bounds.
>
> That said, we agree that the ordering may influence empirical performance, as different orderings can lead to different intermediate decisions during sequential optimization. Investigating adaptive or data-driven ordering strategies could further improve performance in practice, and we view this as an interesting direction for future work.

---

> > ### Author Rebuttal · Reviewer_xJUJ · 2026-04-02
> >
> > Thank you for your response! I will keep my current score

---

### Official Review · Reviewer_K4GV · 2026-03-12

**Soundness:** 4
**Presentation:** 3
**Significance:** 3
**Originality:** 3
**Overall Recommendation:** 4
**Confidence:** 3

**Summary:**

The paper investigates cooperative multi-agent reinforcement learning characterized by a common submodular reward. The authors demonstrate that finding optimal policies in this setting is computationally NP-hard, as the single-step case is equivalent to submodular maximization under a partition matroid constraint. To address this, the paper proposes two primary algorithms: for the known transition function case, it introduces Greedy Policy Optimization, which provides a $1/2$-approximation guarantee with respect to the optimal policy; for the unknown transition function case, it presents UCB-GVI, an algorithm based on the principle of optimism in the face of uncertainty. The authors prove that UCB-GVI achieves a sublinear $1/2$-approximation regret guarantee.

**Compliance With Llm Reviewing Policy:**

Affirmed.

**Key Questions For Authors:**

1. What is the reason for assuming a fixed initial state? Would the theoretical guarantees still hold if the initial state were sampled from a distribution?

2. Is the optimal policy used to establish the $1/2$-approximation guarantee restricted to local policies, or does it also include policies that depend on the joint state?

3. In Lemma 6 and line 891, is a factor of 3 missing in the denominator of the fraction?

**Limitations:**

Yes.

**Strengths And Weaknesses:**

**Strengths**
- The paper is well written and well structured.
- The theorems presented in the paper are sound based on the provided proofs.
- The work successfully addresses the curse of dimensionality by developing algorithms whose complexity scales polynomially with the number of agents $K$.
- By leveraging the submodular structure through sequential greedy optimization, the paper establishes a $1/2$-approximation guarantee relative to the optimal policy under known dynamics.


**Weaknesses**
- The initial state is assumed to be fixed, whereas in most reinforcement learning settings it is typically sampled from an initial state distribution.
- There are some notation-related errors. For example, the definition of $V_h^{\pi}$ in line 180 (below Assumption 1) appears to be incorrect, and $N_{i, h}$ in line 368 is incorrectly introduced.
- There are some minor typographical errors (e.g., a missing closing “}” in Equation (15)).

---

> ### Author Rebuttal · Authors · 2026-03-31
>
> >1.The initial state is assumed to be fixed, whereas in most reinforcement learning settings it is typically sampled from an initial state distribution.
>
> We thank the reviewer for this insightful observation. We adopt a fixed initial state primarily for clarity and to simplify notation, avoiding additional technical overhead that may obscure the main contributions.
>
> Importantly, our results extend directly to the case where the initial state is sampled from a distribution. This can be handled by introducing a dummy initial state $s_0$, which transitions to the true initial state according to the desired distribution, independent of the action. Since this modification adds only a single state and one decision step, it does not affect the order of the computational complexity or the regret guarantees. Therefore, all theoretical results remain unchanged.
>
>
> >2. There are some notation-related errors. For example, the definition of $V_h^{\pi}$ in line 180 (below Assumption 1) appears to be incorrect, and $N_{i,h}$ in line 368 is incorrectly introduced. ...There are some minor typographical errors (e.g., a missing closing “}” in Equation (15)). ...In Lemma 6 and line 891, is a factor of 3 missing in the denominator of the fraction?
>
>  We thank the reviewer for carefully identifying these issues. We will correct the definition of
> $V_h^{\pi}$
> , properly introduce $N_{i,h}$, fix the typographical errors (including Equation (15)), and verify the constants in Lemma 6 and line 891. We will conduct a thorough revision to ensure correctness and clarity in the final version.
>
>
>  * Questions:
>
> >1. What is the reason for assuming a fixed initial state? Would the theoretical guarantees still hold if the initial state were sampled from a distribution?
>  Please see our answer above.
>
>  Please see our response above.
>
>
> > 2. Is the optimal policy used to establish the approximation guarantee restricted to local policies, or does it also include policies that depend on the joint state?
>
> The optimal policy is not restricted to local policies and include policies that depend on the joint state.
>
> This highlights the generality of our results: although our algorithm outputs deterministic policies, the performance guarantees hold with respect to a broader class of policies, including randomized ones (e.g., $\epsilon$-greedy or other stochastic sampling-based strategies).
>  This demonstrates that our method remains competitive even when compared against a strictly more general class of policies beyond deterministic ones.
>
> In particular, we establish that for any  **randomized, non-factorized** policy, our method outputs a policy whose value is guaranteed to achieve at least a constant-factor approximation of $1/2$ of that policy.

---

> > ### Author Rebuttal · Reviewer_K4GV · 2026-04-03
> >
> > I thank the authors for their response! I will keep my current score.

---

### Decision · Program_Chairs · 2026-04-30

**Decision:**

Accept (regular)

**Comment:**

The reviewers appreciate the technical novelty and contribution. The authors are also encouraged to better clarify some of the assumptions and connections with prior work.